# INTERACTION BASED GAUSSIAN WEIGHTING CLUSTERING FOR FEDERATED LEARNING

## ABSTRACT

Federated learning emerged as a decentralized paradigm to train models while securing privacy. However, conventional FL faces data heterogeneity and class imbalance challenges, affecting model performance. In response to these issues, Personalized FL has been developed as an innovative methodology that relies on fine-tuning the distinct local models based on individual training datasets. In this work, we propose a novel PFL method, `FedGWC` (Federated Gaussian Weighting), which groups clients based on their data distribution, allowing training of a more robust and personalized model on the identified clusters. `FedGWC` identifies homogeneous clusters by transforming individual empirical losses to model client interactions with a Gaussian reward mechanism. Additionally, we introduce a new clustering metric for FL to evaluate cluster cohesion with respect to the individual class distribution. Our experiments on benchmark datasets show that `FedGWC` outperforms existing FL algorithms in cluster quality and classification accuracy, validating the efficacy of our approach.

## 1 INTRODUCTION

Federated Learning (FL) has emerged as a promising paradigm for training models on decentralized data while preserving privacy and improving communication efficiency. Unlike traditional machine learning frameworks, FL enables collaborative training between multiple clients without requiring raw data transfer, making it particularly attractive in privacy-sensitive domains (McMahan et al., 2017; Bonawitz et al., 2019). FL was introduced primarily to address two major challenges in decentralized scenarios: ensuring privacy (Kairouz et al., 2021) and reducing communication overhead (Hamer et al., 2020; Asad et al., 2020). In particular, FL algorithms must guarantee *communication efficiency* to reduce the burden associated with the exchange of model updates between clients and the central server, while maintaining strong *privacy guarantees* is also essential, as clients should not expose their private data during the training process.

A fundamental challenge in FL is given by data heterogeneity (Li et al., 2020), specifically in the form of class imbalance within individual clients and non-IID distributions throughout the federation. In this scenario, a single global model often fails to generalize due to clients contributing updates from skewed distributions, leading to degraded performance (Zhao et al., 2018; Caldarola et al., 2022) compared to the centralized counterpart. Furthermore, noisy or corrupted data from some clients can further complicate the learning process (Cao et al., 2020; Zhang et al., 2022), while non-IID data often results in unstable convergence and conflicting gradient updates (Hsieh et al., 2020; Zhao et al., 2018). Despite the introduction of various techniques to mitigate these issues, such as regularization methods (Li et al., 2020), momentum (Mendieta et al., 2022), and control variates (Karimireddy et al., 2020b), statistical heterogeneity remains a critical unsolved problem.

Recently, Personalized FL (PFL) techniques have emerged to address the limitations of learning a single global model for the entire federation, shifting the focus to a more flexible and cooperative strategy (Zhang et al., 2021). These approaches seek to maximize both the shared information between clients and the specific data characteristics of individual clients, allowing the creation of *personalized* models that adapt better to the unique distributions present in the data of each client (T Dinh et al., 2020; Tan et al., 2022; Li et al., 2021; Sun et al., 2021). By doing so, these methods mitigate the effects of data heterogeneity and class imbalance and improve the performance of models on individual client tasks. Among PFL methods, clustering techniques have been employed to group clients with similar data distributions, proposing an effective solution to face client statistical heterogeneity (Ghosh et al., 2020; Sattler et al., 2020). By partitioning clients based on data distributions, clustering-based approaches reduce model divergence within each group, allowing for more homogeneous and targeted model updates.

In this work, we address the challenges of data heterogeneity and class imbalance through a novel PFL clustering-based approach. We propose `FedGWC` (Federated Gaussian Weighting), a method to group clients with similar data distributions into clusters, enabling the training of personalized federated models for each group. Using a Gaussian reward mechanism to form homogeneous clusters, our method builds clusters based on an *interaction matrix*, which encodes the compatibility of all the couples of clients according to their data distribution. This is achieved by simply communicating the empirical losses to the server at each communication round. Each cluster benefits from a personalized federated model that better captures the shared data characteristics within the group, offering a more robust solution to the limitations of training a single global model over heterogeneous data. Training on more homogeneous clusters allows us to exploit the advantages of FL, such as aggregating knowledge from a larger pool of data, while avoiding the drawbacks of statistical heterogeneity, such as client drift. The core idea of our approach is that client data distributions can be inferred by a proper transformation of individual empirical loss functions, motivated by the fact that training federated algorithms on homogeneous clusters leads to better performance compared to training across the entire federation, as shown in (Sattler et al., 2020).

We present a comprehensive mathematical framework for the proposed algorithm and conduct a rigorous analytical examination to establish the convergence properties of the Gaussian weights. Unlike the majority of existing work on FL, we do not rely on model updates to cluster clients. Instead, inspired by (Cho et al., 2022), we extract valuable information from individual losses, hypothesizing that clients with similar data distributions exhibit similar loss landscapes. Furthermore, `FedGWC` can be integrated with any robust FL aggregation algorithm to furtherly handle data heterogeneity. Additionally, we introduce a new clustering metric tailored for evaluating cluster cohesion in the presence of class imbalance. Through experiments on benchmark datasets, we demonstrate that our approach outperforms existing personalization and clustering algorithms in terms of accuracy and clustering quality.

**Contributions.**

- We propose an efficient FL framework that clusters clients based on class imbalance and data heterogeneity, leading to personalized models within clusters.
- We introduce a new clustering metric specifically designed to evaluate clusters in the presence of class imbalance.
- We provide a rigorous mathematical framework to motivate the algorithm, showing its convergence properties.
- We compare `FedGWC` with clustered FL baselines, empirically showing that `FedGWC` provides the best clusters and improves performance in the most heterogeneous scenarios.
- We empirically show how `FedGWC` can be used with any FL aggregation algorithm.
- We investigate the behavior of our algorithm in class and domain imbalance scenarios, showing how our algorithm successfully clusters the clients according to different class distributions or domains.

## 2 RELATED WORK

**FL with Heterogeneous Data.** Handling data heterogeneity, especially class imbalance, remains a critical challenge in FL. `FedProx`(Li et al., 2020) was one of the first attempts to address heterogeneity by introducing a proximal term that constrains local model updates close to the global model. `FedMD` (Li & Wang, 2019) focuses on heterogeneity in model architectures, allowing collaborative training between clients with different neural network structures using model distillation. Methods such as `SCAFFOLD` (Karimireddy et al., 2020b) and `Mime` (Karimireddy et al., 2020a) have also been proposed to reduce client drift by using control variates during the optimization process, which helps mitigate the effects of non-IID data. Furthermore, strategies such as biased client selection (Cho et al., 2022) based on ranking local losses of clients and normalization of updates in `FedNova` (Wang et al., 2021) have been developed to specifically address class imbalance in federated networks, leading to more equitable global model performance.

**Personalization in FL.** Personalization in FL has been extensively explored to tackle the challenge of data heterogeneity across clients (Tan et al., 2022). Various techniques aim to adapt models to each client's specific needs while maintaining a collaborative training framework. A notable method is `FedPer` (Arivazhagan et al., 2019), which personalizes the last layers of the model while sharing the lower layers across all clients. Another important contribution is the `pFedMe` algorithm (T Dinh et al., 2020), which uses a Moreau envelope to decouple local and global model updates,

allowing for client-specific customization without losing the benefits of collaborative learning. Similarly, `Per-FedAvg` (Fallah et al., 2020) leverages a model-agnostic meta-learning (MAML) approach (Finn et al., 2017), optimizing a global model while fine-tuning each client locally, ensuring that the global model can generalize across clients but is also personalized. The `Ditto` framework (Li et al., 2021) further advances personalization by ensuring fairness and robustness, particularly in non-IID settings, through individualized model training. More recently, `FedALA` (Zhang et al., 2023) has been introduced, offering adaptive local aggregation for enhanced personalization.

**Clustering in FL.**  Clustering has proven to be an effective strategy in FL for handling client heterogeneity and improving personalization (Huang et al., 2022; Duan et al., 2021; Briggs et al., 2020; Caldarola et al., 2021; Ye et al., 2023). Clustered FL (Sattler et al., 2020) is one of the first methods proposed to group clients with similar data distributions to train specialized models rather than relying on a single global one. Nevertheless, from a practical perspective, this method exhibits pronounced sensitivity to hyper-parameter tuning, especially concerning the gradient norms threshold, which is intricately linked to the dataset. This sensitivity can result in significant issues of either excessive under-splitting or over-splitting. Additionally, as client sampling is independent of the clustering, there may be privacy concerns due to the potential for updating cluster models with the gradient of a single client. An extension of this is the efficient framework for clustered FL proposed by (Ghosh et al., 2020), which strikes a balance between model accuracy and communication efficiency. Multi-Center FL (Long et al., 2023) builds on this concept by dynamically adjusting client clusters to achieve better personalization, however a-priori knowledge on the number of clusters is needed. Similarly, `IFCA` (Ghosh et al., 2020) addresses client heterogeneity by predefining a fixed number of clusters and alternately estimating the cluster identities of the users by optimizing model parameters for the user clusters via gradient descent. However, it imposes a significant computational burden, as the server communicates all cluster models to each client, which must evaluate every model locally to select the best fit based on loss minimization. This approach not only increases communication overhead but also introduces inefficiencies, as each client must test all models, making it less scalable in larger networks.

Compared to previous approaches, the key advantage of the proposed algorithm, `FedGWC`, lies in its ability to effectively identify clusters of clients with similar levels of heterogeneity and class distribution through simple transformations of individual empirical losses. This is achieved without imposing significant communication overhead or requiring additional computational resources. Additionally, `FedGWC` can be seamlessly integrated with any aggregation method, enhancing its robustness and performance when dealing with heterogeneous scenarios.

## 3  FedGWC: Mathematical Framework

In this section, we introduce the mathematical framework of our algorithm, FEDERATED GAUSSIAN WEIGHTING CLUSTERING (`FedGWC`), a recursive clustered FL algorithm designed to group clients with similar data distributions, where each cluster develops its own model.

First, in Section 3.1 we introduce the mathematical formulation and notations of the FL problem we aim to solve. Then, in Sections 3.2 and 3.3, we present the recursive step of our algorithm or, in other words, how `FedGWC` decides if a cluster needs to be split further into smaller but more homogeneous clusters, or if the current cluster is already homogeneous enough. Finally, Section 3.4 introduces the full algorithmic notation, including cluster indices, and presents the overall recursive structure. To improve clarity, we initially omit cluster indices, focusing instead on the internal mechanics of the recursive step. In particular, without loss of generality, we consider a cluster of clients as the totality of the clients in the federation in Sections 3.2 and 3.3, as the final objective here is to explain how `FedGWC` eventually partitions a set of clients into some smaller clusters of clients. Then, we reintroduce the cluster indices in Section 3.4.

In addition, we propose a novel metric to evaluate the quality of clustered FL algorithms in Section 3.5. This metric, which derives from the Wasserstein distance (Kantorovich, 1942), quantifies the cohesion of client groups based on their class distribution similarities.

### 3.1  Problem Formulation

We adopt the general structure of most FL methods where, during each round of training, a subset of clients downloads the model from the server, trains it locally using its data, and sends back the updated models for future rounds. Let $K$ be the total number of clients and $T$ the total number of communication rounds. At each communication round $t \in [T]$, a subset $\mathcal{P}_t$ of participating clients are selected for training. Let $S$ be the total number of training iterations that are performed at each com-

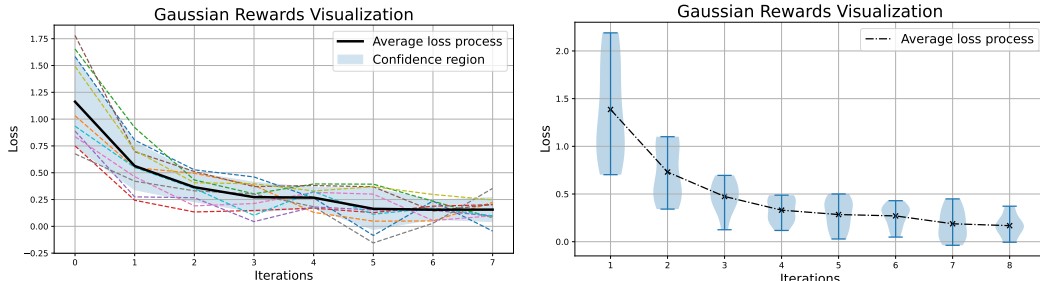

Figure 1: The principle underlying Gaussian reward mechanisms is that if the client's loss lies within the confidence region, the algorithm assigns a higher Gaussian reward. Conversely, if the loss is further from the confidence region, it assigns a lower reward. *Left:* Empirical loss processes , dashed lines, over $S = 8$ local iterations for 10 sampled clients *i.e.* $l_k^{t,s}$ for $k \in \mathcal{P}_t$ and $s = 1, \ldots, S$, the average loss is represented in black, *i.e.* $\hat{\mu}^{t,s}$ for $s = 1, \ldots, S$ and the light blue region delineating the confidence within the standard deviation *i.e.* $\hat{\sigma}^{t,s}$ for $s = 1, \ldots, S$. *Right:* The same representation for the identical clients in the same round is provided with violin plots instead of intervals. Blacked dashed line is the average process of the empirical loss, *i.e.* $\hat{\mu}^{t,s}$, with the associated standard errors, *i.e.* $\hat{\mu}^{t,s} \pm \hat{\sigma}^{t,s}$ for $s = 1, \ldots, S$.

munication round during the training. In general, FL aims to solve an optimization problem that can be stated in the following form (McMahan et al., 2017): $\min_{\theta \in \Theta} \mathcal{L}(\theta) = \min_{\theta \in \Theta} \sum_{k=1}^{K} \frac{n_k}{n} \mathcal{L}_k(\theta)$, where $\mathcal{L}_k(\cdot)$ is the loss function of client $k$ with $n_k$ training samples, and $n = \sum_k n_k$. When $k$ is sampled during round $t$, its local parameters are updated at every iteration $s$ with a stochastic optimizer. For instance, Stochastic Gradient Descent (SGD) has the following update rule: $\theta_k^{t,s+1} = \theta_k^{t,s} - \eta \nabla \mathcal{L}_k(\theta_k^{t,s})$, where $\theta_k^{t,s}$ is the vector of the model parameters, $\eta$ the learning rate. The stochastic process identified by the optimization algorithm suggests that the evolution of the empirical loss can be modeled using random variables and tools from probability theory. In the following sections, we will use capital letters (*e.g.*, $X$) to denote random variables and lowercase letters (*e.g.*, $x$) to represent their specific observations.

### 3.2 GAUSSIAN REWARDS WEIGHTS

To assess how closely each client's local data aligns with the global distribution, we introduce the *Gaussian Weights* $\gamma_k$, statistical estimators that capture the closeness of each clients' class distribution to the main distribution of the cluster. A weight near zero suggests that the client's distribution is far from the main distribution, meaning that it should probably belong to a separate cluster. We pictorially represent the idea of the Gaussian rewards in Figure 1.

The core idea of FedGWC is to group clients based on the similarity of their empirical losses, which are used to compute the *rewards*, continuous random variables in $(0, 1)$. A high reward indicates that a client's loss is close to the cluster's mean loss, while a lower reward reflects greater divergence. Gaussian weights estimate the expected value of these rewards, quantifying the closeness between each client's distribution and the central distribution.

At each training round $t$ and local training iteration $s$, we define the loss random variable $L_k^{t,s}$, of which we observe its samples $l_k^{t,s} = \mathcal{L}_k(\theta_k^{t,s})$. These values are naturally computed in the clients during the local training. For each client $k$, it is possible to define a family of random *rewards* that are stationary by construction, *i.e.*, their moments do not depend on the iteration and are obtained with a Gaussian transformation of the loss

$$R_k^{t,s} = \exp\left(-\frac{(L_k^{t,s} - \hat{\mu}^{t,s})^2}{2(\hat{\sigma}^{t,s})^2}\right), \tag{1}$$

where $\hat{\mu}^{t,s} = 1/|\mathcal{P}_t| \sum_{k \in \mathcal{P}_t} L_k^{t,s}$ is the sample mean across clients, and $(\hat{\sigma}^{t,s})^2 = 1/(|\mathcal{P}_t| - 1) \sum_{k \in \mathcal{P}_t} (L_k^{t,s} - \hat{\mu}^{t,s})^2$ is the sample variance. The rewards $R_k^{t,s} \to 1$ as the distance between $L_k^{t,s}$ and the average $\hat{\mu}^{t,s}$ decreases, while $R_k^{t,s} \to 0$ as the distance increases. Therefore, the expected reward $\mathbb{E}[R_k^{t,s}]$ for out-of-distribution clients is lower than that of the in-distribution clients. In Eq. 1, during any local iteration $s$, a Gaussian kernel with mean loss and sample variance assesses a client's proximity to the confidence interval's center, indicating their probability of sharing the same learning process distribution.

To estimate the expected reward $\mathbb{E}[R_k^{t,s}]$, we introduce the random variable $\Omega_k^t$, which is the average of the rewards across the $S$ local iterations, *i.e.*, $\Omega_k^t = 1/S \sum_{s \in [S]} R_k^{t,s}$. We introduce $\Omega_k^t$ to reduce noise in the estimation caused by stochastic fluctuations in the loss. Instead of using a single sample, such as the last value of the loss as done in Cho et al. (2022), we opted for averaging across iterations to provide a more stable estimate. Due to the linearity of the expectation operator, the expected reward $\mathbb{E}[R_k^{t,s}]$ for the $k$-th client at round $t$, local iteration $s$ equals the expected Gaussian reward $\mathbb{E}[\Omega_k^t]$ that, to simplify the notation, we denote by $\mu_k$. $\mu_k$ is the theoretical value that we aim to estimate by designing our Gaussian weights $\Gamma_k^t$ appropriately, as it encodes the ideal reward to quantify the closeness of the distribution of each client $k$ to the main distribution. Note that the process is stationary by construction. Therefore, it does not depend on $t$ but differs between clients, as it reaches a higher value for in-distribution clients and a lower for out-of-distribution clients.

Practically, we calculate the observed Gaussian weights $\gamma_k^t$. We initialize $\gamma_k^t$ to zero, as we do not want to bias the estimation of the expectation of the reward. During each round $t$, a group of participating clients $\mathcal{P}_t$ is sampled. Each client $k \in \mathcal{P}_t$ updates its model parameters $\theta_k^{t+1}$, stores the observed loss process $l_k^t = (l_k^{t,1}, \ldots, l_k^{t,S}) \in \mathbb{R}^S$ and communicates it to the server, along with the update model parameters $\theta_k^{t+1}$. Then, the server computes the observed reward process $r_k^t = (r_k^{t,1}, \ldots, r_k^{t,S}) \in [0,1]^S$ according to Eq. 1, and $\omega_k^t = 1/S \sum_{s \in [S]} r_k^{t,s}$, *i.e.*, the realization of the random variable $\Omega_k^t$, averaging the entries of $r_k^t$. Finally the server can update the weight $\gamma_k^t$ for each selected client $k \in \mathcal{P}_t$ as

$$\gamma_k^{t+1} = (1 - \alpha_t)\gamma_k^t + \alpha_t \omega_k^t \tag{2}$$

for a sequence of update coefficients $\{\alpha_t\}_{t=0}^\infty$ such that $0 < \alpha_t < 1 \; \forall t$. The weight definition in Eq.2 is closely related to the Robbins-Monro stochastic approximation method (Robbins & Monro, 1951). If a client is not participating in the training, its weight is not updated. FedGWC mitigates biases in the estimation of rewards by employing two mechanisms: (1) uniform random sampling method for clients, with a dynamic adjustment process to prioritize clients that are infrequently sampled, thus ensuring equitable participation across time periods; and (2) when a client is not sampled in a round, its weight and contribution to the reward estimate remain unchanged.

To rigorously motivate the construction of our algorithm and the reliability of the weights, we introduce the following theoretical results. Theorems 3.1 and 3.2 demonstrate that the weights converge to a finite value and, more importantly, that this limit serves as an unbiased estimator of the theoretical reward $\mu_k$. The first theorem provides a strong convergence result, showing that, with suitable choices of the sequence $\{\alpha_t\}_t$, the expectation of the Gaussian weights $\Gamma_k^t$ converges to $\mu_k$ in $L^2$ and almost surely. In addition, Theorem 3.2 extends this to the case of constant $\alpha_t$, proving that the weights still converge and remain unbiased estimators of the rewards as $t \to \infty$.

**Theorem 3.1.** *Let $\{\alpha_t\}_{t=1}^\infty$ be a sequence of positive real values, and $\{\Gamma_k^t\}_{t=1}^\infty$ the sequence of Gaussian weights. If $\{\alpha_t\}_{t=1}^\infty \in l^2(\mathbb{N})/l^1(\mathbb{N})$, then $\Gamma_k^t$ converges in $L^2$. Furthermore, for $t \to \infty$,*

$$\Gamma_k^t \longrightarrow \mu_k \;\; a.s. \tag{3}$$

**Theorem 3.2.** *Let $\alpha \in (0,1)$ be a fixed constant, then in the limit $t \to \infty$, the expectation of the weights converges to the individual theoretical reward $\mu_k$, for each client $k = 1, \ldots, K$, i.e.,*

$$\mathbb{E}[\Gamma_k^t] \longrightarrow \mu_k \;\; t \to \infty \,. \tag{4}$$

Finally, Proposition 3.1 shows that Gaussian weights reduce the variance of the estimate, thus decreasing the error and enabling the construction of a confidence interval for $\mu_k$.

**Proposition 3.1.** *The variance of the weights $\Gamma_k^t$ is smaller than the variance $\sigma_k^2$ of the theoretical rewards $R_k^{t,s}$.*

Complete proofs of Theorems 3.1, 3.2 and Proposition 3.1 are detailed in Appendix A.

### 3.3 INTERACTION MATRIX AND CLUSTERING

**Interaction Matrix.** In the previous section, we defined the Gaussian weights as a measure of proximity between each client's data distribution and the main distribution of the cluster. Gaussian weights are scalar quantities that offer an absolute measure of the alignment between a client's data distribution and the global distribution. Although these weights indicate the conformity of each client's distribution individually, they do not consider the interrelations among the distributions of different clients. Therefore, we propose to encode these interactions in an *interaction matrix* $P^t \in \mathbb{R}^{K \times K}$ whose element $P_{kj}^t$ estimates the similarity between the $k$-th and the $j$-th client data

distribution. The interaction matrix is initialized to the null matrix, *i.e.* $P_{kj}^0 = 0$ for every couple $k, j \in [K]$.

Specifically, we define the update rule for the matrix $P^t$ as follows:

$$P_{kj}^{t+1} = \begin{cases} (1 - \alpha_t) P_{kj}^t + \alpha_t \omega_k^t & , \quad (k, j) \in \mathcal{P}_t \times \mathcal{P}_t \\ P_{kj}^t & , \quad (k, j) \notin \mathcal{P}_t \times \mathcal{P}_t \end{cases} \tag{5}$$

where $\{\alpha_t\}_t$ is the same sequence used to update the weights, and $\mathcal{P}_t$ is the subset of clients sampled in round $t$.

Intuitively, in the long run, since $\omega_k^t$ measures the proximity of the loss process of client $k$ to the average loss process of clients in $\mathcal{P}_t$ at round $t$, we are estimating the *expected perception* of client $k$ by client $j$ with $P_{kj}^t$, *i.e.* a larger value indicates a higher degree of similarity between the loss profiles, whereas smaller values indicate a lower degree of similarity. For example, if $P_{kj}^t$ is close to 1, it suggests that on average, whenever $k$ and $j$ have been simultaneously sampled prior to round $t$, $\omega_k^t$ was high, meaning that the two clients are well-represented within the same distribution.

Interestingly, it is possible to show that the diagonal values of the interaction matrix are exactly the Gaussian weights computed as in Eq. 2. Indeed, by looking at Eq. 5, if we take $k = j$, we obtain the same relation in Eq. 2; namely, for any $k$ and any $t$, the equality $P_{kk}^t = \gamma_k^t$ holds. Moreover, the entries of the interaction matrix $P^t$ are bounded, as shown in Proposition A.2. Furthermore, as a direct consequence of Theorem 3.2, there exists a matrix $P \in \mathbb{R}^{K \times K}$, such that, in the limit for $t \to \infty$, $\mathbb{E}[P_{kj}^t] \to P_{kj}$ entry-wise.

To effectively extract the information embedded in $P$, we introduce the concept of *unbiased perception vectors* (UPV). For any pair of clients $k, j \in [K]$, the UPV $v_k^j \in \mathbb{R}^{K-2}$ represents the $k$-th row of $P$, excluding the $k$-th and $j$-th entries. Recalling the construction of $P^t$, where each row indicates how a client is perceived to share the same distribution as other clients in the federation, the UPV $v_k^j$ captures the collective perception of client $k$ by all other clients, excluding both itself and client $j$. This exclusion is why we refer to $v_k^j$ as *unbiased*.

The UPVs encode information about the relationships between clients, which can be exploited for clustering. However, the UPVs cannot be directly used as their entries are only aligned when considered in pairs. Instead, we construct the *affinity matrix* $W$ by transforming the information encoded by the UPVs through an RBF kernel, as this choice allows to effectively model the affinity between clients: two clients are considered *affine* if similarly perceived by others. This relation is encoded by the entries of $W \in \mathbb{R}^{K \times K}$, which we define as:

$$W_{kj} = \mathcal{K}(v_k^j, v_j^k) = \exp\left(-\beta \left\| v_k^j - v_j^k \right\|^2\right). \tag{6}$$

The spread of the RBF kernel is controlled by a single hyper-parameter $\beta$: changes in this value provide different clustering outcomes, as shown in the sensitivity analysis in Appendix H.

**Clustering.** The affinity matrix $W$, designed to be symmetric, highlights features that capture similarities between clients' distributions. Clustering is performed by the server using the rows of $W$ as feature vectors, as they contain the relevant information. We apply the spectral clustering algorithm (Ng et al., 2001) to $W$ due to its effectiveness in detecting non-convex relationships embedded within the client affinities. Symmetrizing the interaction matrix $P$ into the affinity matrix $W$ is fundamental for spectral clustering as it refines inter-client relationship representation. It models interactions, reducing biases, and emphasizing reliable similarities. This improves robustness to noise, allowing spectral clustering to effectively detect the distributional structure underlying the clients' network (Von Luxburg, 2007). During the iterative training process, the server determines whether to perform clustering by checking the convergence of the matrix $P^t$. Convergence is numerically verified when the mean squared error (MSE) between consecutive updates is less than a small threshold $\epsilon > 0$. To reduce the computational cost of computing the MSE at each round, we employ a running average update. Specifically, the MSE is initialized to 1 in order to ensure stability and avoid erratic updates in early iterations, and it is updated as follows:

$$\text{MSE}^{t+1} = (1 - \alpha_t)\text{MSE}^t + \alpha_t m^t \quad \text{with} \quad m^t = \frac{1}{|\mathcal{P}_t|} \sum_{k, j \in \mathcal{P}_t} |P_{kj}^t - P_{kj}^{t+1}|^2. \tag{7}$$

Algorithm 1 summarizes the clustering procedure. If the MSE is below $\epsilon$, the server computes the matrix $W^t$ and performs spectral clustering over $W^t$ with a number of clusters $n \in \{2, \ldots, n_{max}\}$.

For each clustering outcome, the Davies-Bouldin (DB) score (Davies & Bouldin, 1979) is computed: DB larger than one means that clusters are not well separated, while if it is smaller than one, the clusters are well separated.

We denote by $n_{cl}$ the optimal number of clusters detected by FedGWC. If $\min_{n=2,\ldots,n_{max}} DB_n > 1$, we do not split the current cluster. Hence, the optimal number of clusters is $n_{cl}$ is one. In the other case, the optimal number of clusters is $n_{cl} \in \arg\min_{n=2,\ldots,n_{max}} DB_n$. This requirement ensures proper control over the over-splitting phenomenon, a common issue in hierarchical clustering algorithms in FL. Over-splitting can undermine key principles of FL by creating degenerate clusters with very few clients. Moreover, since each cluster requires a server for aggregation, two options arise: either one client from each cluster acts as the server, or the central server manages the clusters separately. In the first case, this assumption is unrealistic in cross-device settings, as clients typically have limited resources. In the second case, the server would face an excessive workload, along with a significant communication overhead from managing multiple models and performing separate aggregations. For these reasons, over-splitting is particularly problematic in cross-device FL. Finally, on each cluster $\mathcal{C}^{(1)}, \ldots, \mathcal{C}^{(n_{cl})}$ an FL aggregation algorithm is trained separately, resulting in models $\theta_{(1)}, \ldots, \theta_{(n_{cl})}$ personalized for each cluster.

### 3.4 FEDGWC ALGORITHM

In the previous sections, we have detailed the recursive step within the individual clusters. In this section, we present the full FedGWC recursive procedure (Algorithm 2), introducing the complete notation with indices for the distinct clusters. We denote the clustering index as $n$, and the total number of clusters $N_{cl}$.

The interaction matrix $P_{(1)}^0$ is initialized to the null matrix $0_{K \times K}$, and the *total number of clusters* $N_{cl}^0$, as no clusters have been formed yet, and $\text{MSE}_{(1)}^0$ are initialized to 1, in order to ensure stability in early updates, allowing a gradual decrease. At each communication round $t$, and for cluster $\mathcal{C}^{(n)}$, where $n = 1, \ldots, N_{cl}^t$, the cluster server independently samples the participating clients $\mathcal{P}_t^{(n)} \subseteq \mathcal{C}^{(n)}$. Each client $k \in \mathcal{P}_t^{(n)}$ receives the current cluster model $\theta_{(n)}^t$. After performing local updates, each client sends its updated model $\theta_k^{t+1}$ and empirical loss $l_k^t$ back to the cluster server. The server aggregates these updates to form the new cluster model $\theta_{(n)}^{t+1}$, computes the Gaussian rewards $\omega_k^t$ for the sampled clients, and updates the interaction matrix $P_{(n)}^{t+1}$ and $\text{MSE}_{(n)}^{t+1}$ according to Eqs. 5 and 7. If $\text{MSE}_{(n)}^{t+1}$ is lower than a threshold $\epsilon$, the server of the cluster performs clustering to determine whether to split cluster $\mathcal{C}^{(n)}$ into $n_{cl}$ sub-groups, as outlined in Algorithm 1. The matrix $P_{(n)}^{t+1}$ is then partitioned into sub-matrices by filtering its columns and rows according to the newly formed clusters, with the MSE for these sub-matrices reinitialized to 1. This process results in a distinct model $\theta_{(n)}$ for each cluster $\mathcal{C}^{(n)}$. When the final iteration $T$ is reached we are left with $N_{cl}^T$ clusters with personalized models $\theta_{(n)}$ for $n = 1, \ldots, N_{cl}^T$.

Thanks to the Gaussian Weights, and the recursive spectral clustering on the affinity matrices, our algorithm, FedGWC, is able to detect groups of clients that display similar levels of heterogeneity. The clusters formed are more uniform, *i.e.* the class distributions within each group are more similar. These results are supported by experimental evaluations, discussed in Section 4.2.

### 3.5 A NEW METRIC TO EVALUATE CLUSTERING IN FL

In the previous section we observed that when clustering clients according to different heterogeneity levels, the outcome must be evaluated using a metric that assesses the cohesion of individual distributions. In this paragraph, we introduce a novel metric to evaluate the performance of clustering algorithms in FL. This metric, derived the Wasserstein distance (Kantorovich, 1942), quantifies the cohesion of client groups based on their class distribution similarities.

We propose a general method for adapting clustering metrics to account for class imbalances. This adjustment is particularly relevant when the underlying class distributions across clients are skewed. The formal derivation and mathematical details of the proposed metric are provided in Appendix B. In this section, we provide a high-level overview of our new metric.

Consider a generic clustering metric $s$, e.g. Davies-Bouldin score (Davies & Bouldin, 1979) or the Silhouette score (Rousseeuw, 1987). Let $C$ denote the total number of classes, and $x_i^k$ the empirical frequency of the $i$-th class in the $k$-th client's local training set. Following theoretical reasonings, as shown in Appendix B, the empirical frequency vector for client $k$, denoted by $x_{(i)}^k$, is ordered

according to the rank statistic of the class frequencies, *i.e.* $x_{(i)}^k \geq x_{(i+1)}^k$ for any $i = 1, \ldots, C-1$. The class-adjusted clustering metric $\tilde{s}$ is defined as the standard clustering metric $s$ computed on the ranked frequency vectors $x_{(i)}^k$. Specifically, the distance between two clients $j$ and $k$ results in

$$\frac{1}{C} \left( \sum_{i=1}^{C} \left| x_{(i)}^k - x_{(i)}^j \right|^2 \right)^{1/2} . \tag{8}$$

This modification ensures that the clustering evaluation is sensitive to the distributional characteristics of the class imbalance. As we show in Appendix B, this adjustment is mathematically equivalent to assessing the dispersion between the empirical class probability distributions of different clients using the Wasserstein distance, also known as the Kantorovich-Rubenstein metric (Kantorovich, 1942). This equivalence highlights the theoretical soundness of using ranked class frequencies to better capture the variation in class distributions when evaluating clustering outcomes in FL.

## 4 EXPERIMENTS

In this section, we present the experimental results on widely used FL benchmark datasets (Caldas et al., 2018), comparing the performance of `FedGWC` with other baselines from the literature, including standard FL algorithms, personalized FL approaches, and clustering methods. The details on the dataset we use and the implementation choices are shown in Appendix G.

In Section 4.1, we first evaluate our method, `FedGWC`, against various clustering algorithms, including `CFL` (Sattler et al., 2020), `FeSEM` (Long et al., 2023), and `IFCA` (Ghosh et al., 2020), to demonstrate that `FedGWC` yields superior and more consistent clustering outcomes in heterogeneous scenarios than the baselines. Then, we investigate the benefits of integrating `FedGWC` with established FL baselines. Specifically, we test our approach with `FedAvg` (McMahan et al., 2017), `FedAvgM` (Asad et al., 2020) and `FedProx` (Li et al., 2020), showing how our approach is orthogonal to conventional FL aggregation methods. Finally, we compare `FedGWC` with `FeSEM` when paired with personalized FL algorithms such as `pFedMe` (T Dinh et al., 2020) and `Per-FedAvg` (Fallah et al., 2020), showing that our proposed solution exhibits greater robustness than existing clustering techniques and can be effectively combined with pure personalization FL techniques to achieve improved performance over these baselines. Finally, in Section 4.2, we propose analyses on class and domain imbalance, showing that our algorithm successfully detects clients belonging to separate distributions. Further experiments are presented in Appendix J.

### 4.1 FEDGWC IN HETEROGENEOUS SETTINGS

**FedGWC vs. clustering baselines.** We assess the performance of `FedGWC` in comparison to several FL clustering baselines (`IFCA`, `FeSEM`, and `CFL`) utilizing `FedAvg` aggregation. For the algorithms that require knowing the number of clusters in advance, *i.e.* `FeSEM` and `IFCA`, we show only the best result among 2, 3, 4, and 5 clusters. The complete tuning is shown in Appendix I. While `IFCA` showcases competitive outcomes, it incurs significant communication overhead, as each client is required to evaluate models from every cluster in each round, rendering this approach impractical in cross-device scenarios and positioning it as an upper bound in our analysis. Although `FeSEM` is less costly, it is constrained by a predefined number of clusters, reducing flexibility. On the other hand, `CFL` demands extensive hyperparameter tuning – in contrast, `FedGWC` requires only one hyperparameter – and produces overly granular clustering, resulting in an unrealistic number of models for cross-device scenarios, or no clusters at all.

In Table 1, we present a comparative analysis of these algorithms with respect to balanced accuracy, adjusted silhouette score (AS), and adjusted Davies-Bouldin index (ADB), employing `FedAvg` as the aggregation strategy. Recall that higher the value of AS the better the clustering outcome, as, for ADB, a lower value suggests a better cohesion between clusters.

Notably, both `FedGWC` and `CFL` autonomously determine the optimal number of clusters based on data heterogeneity, thereby offering a more scalable solution for large-scale cross-device FL. In contrast to `CFL`, `FedGWC` consistently produced a reasonable number of clusters, even when using the optimal hyperparameters for `CFL`, which resulted in no splits, thereby achieving performance equivalent to FedAvg. Interestingly, as illustrated in Figure 2, `FedGWC` exhibits a significant improvement in accuracy on Cifar100 precisely at the rounds where clustering occurs. Furthermore, as shown in Table 1, `FedGWC` consistently outperformed other methods on both Cifar10 and Cifar100 when evaluated using adjusted clustering metrics, indicating its superior ability to partition clients into homogeneous clusters.

Table 1: Clustered FL baselines: `FedGWC` consistently outperforms the baselines in the most heterogeneous settings (Cifar100 and Femnist), even surpassing the `IFCA` upper-bound.

|  | | Clustering method | C | Acc | AS | ADB |
|---|---|---|---|---|---|---|
| **Dataset** | Cifar10 | IFCA | 2 | **78.6** ± **2.3** | 0.0 ± 0.0 | 17.1 ± 8.1 |
| | | FeSem | 3 | 71.5 ± 1.3 | -0.1 ± 0.0 | 52.7 ± 29.9 |
| | | CFL | 1 | 76.2 ± 0.9 | / | / |
| | | FedGWC | 3 | 75.8 ± 1.1 | **0.1** ± **0.0** | **2.6** ± **0.0** |
| | Cifar100 | IFCA | 5 | 47.5 ± 3.5 | -0.8 ± 0.2 | 5.2 ± 5.1 |
| | | FeSem | 5 | 53.4 ± 1.8 | -0.3 ± 0.1 | 38.4 ± 13.0 |
| | | CFL | 1 | 41.6 ± 1.3 | / | / |
| | | FedGWC | 4 | **53.4** ± **0.4** | **0.1** ± **0.0** | **2.4** ± **0.4** |
| | Femnist | IFCA | 5 | **76.7** ± **0.6** | **0.3** ± **0.1** | **0.5** ± **0.1** |
| | | FeSem | 2 | 75.6 ± 0.2 | 0.0 ± 0.0 | 25.6 ± 7.8 |
| | | CFL | 1 | 76.0 ± 0.1 | / | / |
| | | FedGWC | 4 | 76.1 ± 0.1 | -0.2 ± 0.1 | 18.0 ± 6.2 |

Figure 2: Balanced accuracy on Cifar100 for `FedGWC` using `FedAvg` aggregation compared to the clustered FL baselines. `FedGWC` detects two splits, and demonstrates significant improvements in accuracy when clustering is performed. `FedGWC` has a faster and more stable convergence with respect to baseline algorithms.

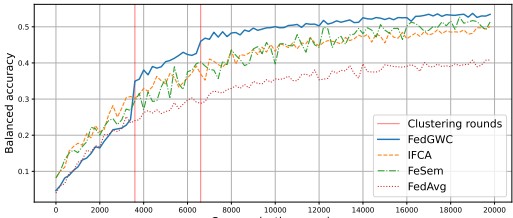

Table 2: Comparison of classical FL algorithms using `FedGWC` vs. the same algorithms without `FedGWC`. Our empirical results suggest that employing our clustering algorithm provides benefits in most heterogeneous settings, such as Cifar100 and Femnist, while it is unnecessary in simpler settings like Cifar10.

| FL method | Cifar10 | | Cifar100 | | Femnist | |
|---|---|---|---|---|---|---|
| | no clusters | FedGWC | no clusters | FedGWC | no clusters | FedGWC |
| FedAvg | **76.2** ± **0.9** | 75.8 ± 1.1 | 41.6 ± 1.3 | **53.4** ± **0.4** | 76.0 ± 0.1 | **76.1** ± **0.1** |
| FedAvgM | **78.6** ± **1.3** | 77.8 ± 2.0 | 41.5 ± 0.5 | **50.5** ± **0.3** | **83.3** ± **0.3** | 83.2 ± 0.4 |
| FedProx | **76.1** ± **0.1** | 75.6 ± 0.8 | 41.8 ± 1.0 | **49.1** ± **1.0** | 75.9 ± 0.2 | **76.3** ± **0.2** |

**Training FL methods with `FedGWC`.** In this paragraph, we empirically show how `FedGWC` can be used on top of any FL aggregation algorithm. Although `FedGWC` does not improve the results on Cifar10 because of the simplicity of the task, our method consistently improved the performance of FL algorithms for the more heterogeneous settings of Cifar100 and Femnist. We show these results in Table 2. `FedGWC` consistently boosted balanced accuracies across all methods. Notably, on Cifar100, the scenario that most resembles cross-device FL, `FedGWC` substantially improves the performance, leveraging the increased heterogeneity to construct more homogeneous clusters. This resulted in an average increase of over 10% in balanced accuracy on Cifar100. While the improvement on Femnist was less pronounced, it remained beneficial in most cases, highlighting `FedGWC`'s adaptability to less heterogeneous environments without introducing overhead. These experiments underscore the orthogonal nature of `FedGWC` to FL aggregation strategies, demonstrating its ability to enhance the performance of classical FL algorithms.

**`FedGWC` improves performances of PFL algorithms.** To evaluate the benefits of combining pure personalization with cluster-wise personalization, we conducted a comparative analysis of `FedGWC` with `FeSEM` and the PFL baselines, `pFedMe` and `Per-FedAvg`. Given the cross-device nature of our algorithm, we focused on `FeSEM` as a representative clustering baseline. While PFL methods can be considered upper bounds, as they perform a more fine-grained personalization than clustered methods at the cost of more client resources, our experiments demonstrate that incorporating `FedGWC` with PFL methods can further enhance their performance when personalization is feasible. However, personalization entails higher computational overhead and diverges from the fundamental philosophy of `FedGWC`, which emphasizes grouping clients based on distribution similarity rather than optimizing individual models of the clients. As illustrated in Table 3, `FedGWC` consistently outperforms `FeSEM` and surpasses the pure personalization performance bound in all benchmark datasets when combined with `pFedME` and `Per-FedAvg`, achieving a notable 4.5% improvement on Cifar100.

## 4.2 Analysis on the clustering decisions of `FedGWC`

**`FedGWC` detects different client class distributions.** Here, we investigate the underlying mechanisms behind `FedGWC`'s clustering decisions in heterogeneous scenarios. Specifically, we explore how the algorithm identifies and groups clients based on the non-IID nature of their data distributions, represented by the Dirichlet concentration parameter $\alpha$. For the Cifar-10 dataset, we propose three distinct client partitions: (1) 90 clients with $\alpha = 0$ and 10 clients with $\alpha = 100$; (2) 90 clients with $\alpha = 0.05$ and 10 clients with $\alpha = 100$; and (3) 40 clients with $\alpha = 100$, 30 clients with $\alpha = 0.05$, and 30 clients with $\alpha = 0$. Similarly, for the Cifar-100 dataset, we apply a similar splitting approach, obtaining the following partitions: (1) 100 clients with $\alpha = 0$ and 10 clients with $\alpha = 1000$; (2) 90 clients with $\alpha = 0.5$ and 10 clients with $\alpha = 1000$; and (3) 40 clients with

Table 3: Comparison of PFL algorithms with and without `FedGWC` or `FeSEM` clustering. Empirical results indicate that `FedGWC` outperforms `FeSEM` and enhances the performance of `pFedME` and `per-FedAvg` when integrated with PFL aggregations.

| FL method | Cifar10 | | | Cifar100 | | | Femnist | | |
|---|---|---|---|---|---|---|---|---|---|
| | no clusters | FedGWC | FeSEM | no clusters | FedGWC | FeSEM | no clusters | FedGWC | FeSEM |
| FedAvg | **76.2** $\pm$ **0.9** | 75.8 $\pm$ 1.1 | 71.5 $\pm$ 1.3 | 41.6 $\pm$ 1.3 | **53.4** $\pm$ **0.4** | 53.4 $\pm$ 1.8 | 76.0 $\pm$ 0.1 | **76.1** $\pm$ **0.1** | 75.4 $\pm$ 0.5 |
| pFedME | **93.4** $\pm$ **0.3** | 93.6 $\pm$ 0.1 | 93.4 $\pm$ 0.2 | 89.0 $\pm$ 0.1 | **93.5** $\pm$ **0.1** | 93.3 $\pm$ 0.1 | 71.2 $\pm$ 0.2 | **72.0** $\pm$ **0.2** | 34.5 $\pm$ 0.9 |
| per-FedAvg | **93.1** $\pm$ **0.1** | 93.4 $\pm$ 0.1 | 93.3 $\pm$ 0.1 | 93.4 $\pm$ 0.1 | **93.5** $\pm$ **0.1** | 93.3 $\pm$ 0.1 | 63.6 $\pm$ 0.3 | **63.9** $\pm$ **0.2** | 63.6 $\pm$ 0.2 |

Table 4: Clustering with three different splits on CIFAR-10 and CIFAR-100 datasets. `FedGWC` has superior clustering quality across different splits.

| Dataset | Split | Clustering method | C | AS | ADB |
|---|---|---|---|---|---|
| CIFAR-10 | (10, 0, 90) | IFCA | 2 | / | / |
| | | FeSem | 3 | -0.0 $\pm$ 0.1 | 12.0 $\pm$ 2.0 |
| | | FedGWC | 3 | **0.1** $\pm$ **0.0** | **0.2** $\pm$ **0.0** |
| | (10, 90, 0) | IFCA | 2 | / | / |
| | | FeSem | 3 | -0.0 $\pm$ 0.0 | 12.0 $\pm$ 2.0 |
| | | FedGWC | 3 | **0.2** $\pm$ **0.0** | **0.6** $\pm$ **0.0** |
| | (40, 30, 30) | IFCA | 2 | -0.2 $\pm$ 0.0 | **1.0** $\pm$ **0.0** |
| | | FeSem | 3 | 0.1 $\pm$ 0.1 | 20.6 $\pm$ 7.1 |
| | | FedGWC | 3 | **0.6** $\pm$ **0.1** | **1.0** $\pm$ **0.4** |
| CIFAR-100 | (10, 0, 100) | IFCA | 5 | -0.9 $\pm$ 0.0 | 1.8 $\pm$ 0.0 |
| | | FeSem | 5 | -0.8 $\pm$ 0.2 | 2.6 $\pm$ 0.6 |
| | | FedGWC | 5 | **0.1** $\pm$ **0.1** | **0.2** $\pm$ **0.2** |
| | (10, 90, 0) | IFCA | 5 | -0.0 $\pm$ 0.0 | **5.6** $\pm$ **1.5** |
| | | FeSem | 5 | 0.2 $\pm$ 0.1 | 12.0 $\pm$ 2.0 |
| | | FedGWC | 5 | **0.4** $\pm$ **0.1** | 6.4 $\pm$ 2.0 |
| | (40, 30, 30) | IFCA | 5 | -0.2 $\pm$ 0.0 | 1.0 $\pm$ 0.0 |
| | | FeSem | 5 | -0.2 $\pm$ 0.0 | 33.2 $\pm$ 0.0 |
| | | FedGWC | 3 | **0.4** $\pm$ **0.2** | **0.9** $\pm$ **0.1** |

Table 5: Clustering performance of `FedGWC` on federations with clients from distinct domains, consisting of clean, noisy, and blurred images across CIFAR-10 and CIFAR-100 datasets. Performance is measured using the Rand Index score (Rand, 1971).It is noted that a Rand Index value approaching 1 signifies a perfect correspondence between the clustering ground truth and the assigned labels. It is important to observe that a Rand Index of 1 represents the maximum value this index can reach, and in four instances, `FedGWC` successfully distinguishes all visual domains.

| Dataset | Domain configuration | C | Rand Index |
|---|---|---|---|
| CIFAR-10 | (50, 0, 50) | 2 | 1.0 $\pm$ 0.0 |
| | (50, 50, 0) | 2 | 1.0 $\pm$ 0.0 |
| | (40, 30, 30) | 4 | 0.9 $\pm$ 0.0 |
| CIFAR-100 | (50, 0, 50) | 2 | 1.0 $\pm$ 0.0 |
| | (50, 50, 0) | 2 | 1.0 $\pm$ 0.0 |
| | (40, 30, 30) | 4 | 0.6 $\pm$ 0.0 |

$\alpha = 1000$, 30 clients with $\alpha = 0.05$, and 30 clients with $\alpha = 0$. Unlike `FeSem` and `IFCA`, `FedGWC` is able to effectively detect varying levels of heterogeneity, as demonstrated by the adjusted Silhuette and Davies-Bouldin reported in Table 4, proving its ability to separate clients according to their data distributions.

**`FedGWC` detects different visual client domains.** Here, we focus on scenarios with nearly uniform class imbalance (high $\alpha$ values) but with different visual domains to investigate how `FedGWC` forms clusters in such settings. We incorporated various artificial domains (non-perturbed, noisy, and blurred images) into CIFAR-10 and CIFAR-100 datasets under homogeneous conditions ($\alpha = 100.00$). Our results demonstrate that `FedGWC` effectively clustered clients according to these distinct domains. Table 5 presents the Rand-Index scores, which assess clustering quality based on known domain labels. The high Rand-Index scores, often approaching the upper bound of 1, indicate that `FedGWC` successfully separated clients into distinct clusters corresponding to their respective domains. Figure 8 in the appendix visualizes the interaction matrix $P$, affinity matrix $W$, and the scatter plot of the clustering with respect to the spectral bi-dimensional embedding on CIFAR-10 for the (40, 30, 30) configuration. This anaylsis suggests that `FedGWC` may be applicable for detecting malicious clients in FL, pinpointing a potential direction for future research.

## 5 CONCLUSIONS

We propose `FedGWC`, an efficient clustering algorithm for heterogeneous FL settings addressing the challenge of non-IID data and class imbalance. Unlike existing clustered FL methods, `FedGWC` groups clients by data distributions with flexibility and robustness, simply using the information encoded by the individual empirical loss. `FedGWC` successfully detects homogeneous clusters, as proved by our proposed novel Wasserstein Adjusted Metric. `FedGWC` detects splits by removing out-of-distribution clients, thus simplifying the learning task within clusters without increasing communication overhead or computational cost. Empirical evaluations show that separately training classical FL algorithms on the homogeneous clusters detected by `FedGWC` consistently improves the performance. Additionally, `FedGWC` excels over other clustering techniques in grouping clients uniformly with respect to class imbalance and heterogeneity levels, which is crucial to mitigate the effect of non-IIDness FL. Finally, clustering on different class unbalanced and domain unbalanced scenarios, which are correctly detected by `FedGWC` (see Section 4.2), suggests that `FedGWC` can also be applied to anomaly client detection and to enhance robustness against malicious attacks in future research.

## 6 REPRODUCIBILITY STATEMENT

The reproducibility of the results and the theoretical contributions of this work has been a paramount concern throughout the entire development of this project and while drafting this manuscript. In this section, we provide details and a concise guide to reproduce our results and verify our contributions.

- **Code Availability:** All the code used in our experiments has been included in the supplementary material of the submission. Additionally, we will release online a well-documented and structured final version of the code to allow for easy reproduction of the experiments detailed in the paper.

- **Datasets and data split:** The datasets used in our experiments are publicly available and can be downloaded online. Detailed instructions on accessing and preprocessing the datasets will be provided, along with the final code release. The data splits can be generated directly through our provided code. This ensures that others can replicate our work's exact federated learning scenarios.

- **Theoretical Results:** We provide complete proofs for all theorems and propositions presented in the paper in Appendices A and B.

We are confident that with these resources, all experimental and theoretical results can be reproduced by the community.

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

## A  THEORETICAL RESULTS FOR FEDGWC

This section provides algorithms, in pseudo-code, to describe FedGWC (see Algorithms 1 and 2). Additionally, here we provide the proofs for the convergence results introduced in Section 3, specifically addressing the convergence (Theorems 3.1 and 3.2) and the formal derivation on the variance bound of the Gaussian weights (Proposition 3.1). In addition, we also present a sufficient condition, under which is guaranteed that the overall sampling rate of the training algorithm does not increase and remain unchanged during the training process (Theorem A.3).

**Theorem A.1.** *Let $\{\alpha_t\}_{t=1}^{\infty}$ be a sequence of positive real values, and $\{\Gamma_k^t\}_{t=1}^{\infty}$ the sequence of Gaussian weights. If $\{\alpha_t\}_{t=1}^{\infty} \in l^2(\mathbb{N})/l^1(\mathbb{N})$, then $\Gamma_k^t$ converges in $L^2$. Furthermore, for $t \to \infty$,*

$$\Gamma_k^t \longrightarrow \mu_k \ \ a.s. \tag{9}$$

*Proof.* At each communication round, we compute the samples $r_i^{t,s}$ from $R_k^{t,s}$ via a Gaussian transformation of the observed loss in Eq. 1. Notice that, due to the linearity of the expectation operator, $\mathbb{E}[\Omega_k^t] = \mu_k$, that is the true, unknown, expected reward. The observed value for the random variable is given by $\omega_k^t = 1/S \sum_{s=1}^{S} r_k^{t,s}$, which is sampled from a distribution centered on $\mu_k$. Each client's weight is updated according to

$$\gamma_k^{t+1} = (1 - \alpha_t)\gamma_t + \alpha_t \omega_k^t \ . \tag{10}$$

Since such an estimator follows a Robbins-Monro algorithm, it is proved to converge in $L^2$. In addition, $\Gamma_k^t$ converges to the expectation $\mathbb{E}[\Omega_k^t] = \mu_k$ with probability 1, provided that $\alpha_t$ satisfies $\sum_{t \geq 1} |\alpha_t| = \infty$, and $\sum_{t \geq 1} |\alpha_t|^2 < \infty$ (Harold et al., 1997). □

**Theorem A.2.** *Let $\alpha \in (0,1)$ be a fixed constant, then in the limit $t \to \infty$, the expectation of the weights converges to the individual theoretical reward $\mu_k$, for each client $k = 1, \ldots, K$, i.e.,*

$$\mathbb{E}[\Gamma_k^t] \longrightarrow \mu_k \ \ t \to \infty \ . \tag{11}$$

*Proof.* Recall that $\gamma_k^{t+1} = (1 - \alpha)\gamma_k^t + \alpha \omega_k^t$, where $\omega_k^t$ are samples from $\Omega_k^t$. If we substitute backward the value of $\gamma_k^t$ we can write

$$\gamma_k^{t+1} = (1 - \alpha)^2 \gamma_k^{t-1} + \alpha \omega_k^t + \alpha(1 - \alpha)\omega_k^{t-1} \ . \tag{12}$$

By iterating up to the initialization term $\gamma_k^0$ we get the following formulation:

$$\gamma_k^{t+1} = (1 - \alpha)^{t+1}\gamma_k^0 + \sum_{\tau=0}^{t} \alpha(1 - \alpha)^\tau \omega_k^{t-\tau} \ . \tag{13}$$

Since $\omega_k^t$ are independent and identically distributed samples from $\Omega_k^t$, with expected value $\mu_k$, then the expectation of the weight at the $t$-th communication round would be

$$\mathbb{E}[\Gamma_k^t] = \mathbb{E}\left[(1 - \alpha)^t \gamma_k^0 + \sum_{\tau=0}^{t} \alpha(1 - \alpha)^\tau \Omega_k^{t-\tau-1}\right] \ , \tag{14}$$

that, due to the linearity of expectation, becomes

$$\mathbb{E}[\Gamma_k^t] = (1 - \alpha)^t \gamma_k^0 + \sum_{\tau=0}^{t} \alpha(1 - \alpha)^\tau \mu_k \ . \tag{15}$$

If we compute the limit

$$\lim_{t \to \infty} \mathbb{E}[\Gamma_k^t] = \lim_{t \to \infty} (1 - \alpha)^t \gamma_k^0 + \sum_{\tau=0}^{\infty} \alpha(1 - \alpha)^\tau \mu_k \ , \tag{16}$$

and since $\alpha \in (0,1)$, the first term tends to zero, and also the geometric series converges. Therefore, the expectation of the weights converges to $\mu_k$, namely

$$\lim_{t \to \infty} \mathbb{E}[\Gamma_k^t] = \mu_k \ . \tag{17}$$

□

**Proposition A.1.** *The variance of the weights $\Gamma_k^t$ is smaller than the variance $\sigma_k^2$ of the theoretical rewards $R_k^{t,s}$.*

*Proof.* From Eq.13, we can show that $\mathbb{V}ar(\Gamma_k^t)$ converges to a value that depends on $\alpha$ and the number of local training iterations $S$. Indeed

$$
\mathbb{V}ar(\Gamma_k^t) = \mathbb{V}ar\left((1-\alpha)^t\gamma_k^0 + \sum_{\tau=0}^{t}\alpha(1-\alpha)^\tau\Omega_k^{t-\tau-1}\right)
$$
$$
= \sum_{\tau=0}^{t}\alpha^2(1-\alpha)^{2\tau}\mathbb{V}ar(\Omega_k^t) = \frac{1}{S}\sum_{\tau=0}^{t}\alpha^2(1-\alpha)^{2\tau}\sigma_k^2
$$
(18)

since $\Omega_k^t = 1/S\sum_{s=1}^{S}R_k^{t,s}$.

If we compute the limit, that exists finite due to the hypothesis $\alpha \in (0,1)$, we get

$$
\lim_{t\to\infty}\mathbb{V}ar(\Gamma_k^t) = \frac{\alpha^2\sigma_k^2}{S}\sum_{\tau=0}^{\infty}(1-\alpha)^{2\tau} = \frac{\alpha}{2-\alpha}\frac{\sigma_k^2}{S} < \frac{\sigma_k^2}{S} < \sigma_k^2 \;.
$$
(19)

$\square$

We further demonstrate that the interaction matrix $P^t$ identified by `FedGWC` is entry-wise bounded from above, as established in the following proposition.

**Proposition A.2.** *The entries of the interaction matrix $P^t$ are bounded from above, namely for any $t \geq 0$ there exists a positive finite constant $C_t > 0$ such that*

$$
P_{kj}^t \leq C_t \;.
$$
(20)

*And furthermore*

$$
\lim_{t\to\infty}C_t = 1 \;.
$$
(21)

*Proof.* Without loss of generality we assume that every client of the federation is sampled, and we assume that $\alpha_t = \alpha \in (0,1)$ for any $t \geq 0$. We recall, from Eq.5, that for any couple of clients $k, j \in \mathcal{P}_t$ the entries of the interaction matrix are updated according to

$$
P_{kj}^{t+1} = (1-\alpha)P_{kj}^t + \alpha\omega_k^t \;.
$$
(22)

If we iterate backward until $P_{kj}^0$, we obtain the following update

$$
P_{kj}^{t+1} = (1-\alpha)^{t+1}P_{kj}^0 + \sum_{\tau=0}^{t}\alpha(1-\alpha)^\tau\omega_k^{t-\tau} \;.
$$
(23)

We know that, by constructions, the Gaussian rewards $\omega_k^t < 1$ at any time $t$, therefore the following inequality holds

$$
P_{kj}^t = (1-\alpha)^tP_{kj}^0 + \sum_{\tau=0}^{t}\alpha(1-\alpha)^\tau\omega_k^{t-\tau-1} \leq (1-\alpha)^tP_{kj}^0 + \sum_{\tau=0}^{t}\alpha(1-\alpha)^\tau \;.
$$
(24)

At any round $t$ we can define the costant $C_t$, as

$$
C_t := (1-\alpha)^tP_{kj}^0 + \alpha\sum_{\tau=0}^{t}(1-\alpha)^\tau = (1-\alpha)^tP_{kj}^0 + 1 - (1-\alpha)^{t+1} < \infty \;.
$$
(25)

Moreover, since $\alpha \in (0,1)$, by taking the limit we prove that

$$
\lim_{t\to\infty}C_t = \lim_{t\to\infty}(1-\alpha)^tP_{kj}^0 + 1 - (1-\alpha)^{t+1} = 1 \;.
$$
(26)

$\square$

---

**Algorithm 1** FedGW_Cluster

---

1: **Input:** $P, n_{max}, \mathcal{K}(\cdot, \cdot)$
2: **Output:** cluster labels $y_{n_{cl}}$, and number of clusters $n_{cl}$
3: Extract UPVs $v_k^j, v_j^k$ from $P$ for any $k, j$
4: $W_{kj} \leftarrow \mathcal{K}(v_k^j, v_j^k)$ for any $k, j$
5: **for** $n = 2, \ldots, n_{max}$ **do**
6:      $y_n \leftarrow$ Spectral_Clustering$(W, n)$
7:      $DB_n \leftarrow$ Davies_Bouldin$(W, y_n)$
8:      **if** $\min_n DB_n > 1$ **then**
9:          $n_{cl} \leftarrow 1$
10:     **else**
11:         $n_{cl} \leftarrow \arg\min_n DB_n$
12:     **end if**
13: **end for**

---

**Algorithm 2** FedGWC

---

1: **Input:** $K, T, S, \alpha_t, \epsilon, |\mathcal{P}_t|, \mathcal{K}$
2: **Output:** $\mathcal{C}^{(1)}, \ldots, \mathcal{C}^{(N_{cl})}$ and $\theta_{(1)}, \ldots, \theta_{(N_{cl})}$
3: Initialize $N_{cl}^0 \leftarrow 1$
4: Initialize $P_{(1)}^0 \leftarrow 0_{K \times K}$
5: Initialize $\text{MSE}_{(1)}^0 \leftarrow 1$
6: **for** $t = 0, \ldots, T - 1$ **do**
7:      $\Delta N^t \leftarrow 0$ for each iterations it counts the number of new clusters that are detected
8:      **for** $n = 1, \ldots, N_{cl}^t$ **do**
9:          Server samples $\mathcal{P}_t^{(n)} \in \mathcal{C}^{(n)}$ and sends the current cluster model $\theta_{(n)}^t$
10:        Each client $k \in \mathcal{P}_t^{(n)}$ locally updates $\theta_k^t$ and $l_k^t$, then sends them to the server
11:        $\omega_k^t \leftarrow$ Gaussian_Rewards$(l_k^t, \mathcal{P}_t^{(n)})$, Eq. 1
12:        $\theta_{(n)}^{t+1} \leftarrow$ FL_Aggregator$(\theta_k^t, \mathcal{P}_t^{(n)})$
13:        $P_n^{t+1} \leftarrow$ Update_Matrix$(P_n^t, \omega_k^t, \alpha_t, \mathcal{P}_t^{(n)})$, according to Eq. 5
14:        Update $\text{MSE}_{(n)}^{t+1}$, according to Eq. 7
15:        **if** $\text{MSE}_n^{t+1} < \epsilon$ **then**
16:           Perform FedGW_Cluster$(P_{(n)}^{t+1}, n_{max}, \mathcal{K})$ on $\mathcal{C}^{(n)}$, providing $n_{cl}$ sub-clusters
17:           Update the number of new clusters $\Delta N^t \leftarrow \Delta N^t + n_{cl} - 1$ u
18:           Cluster server splits $P_{(n)}^{t+1}$ filtering rows and columns according to the new clusters
19:           Re-initialize MSE for new clusters to 1
20:        **end if**
21:      **end for**
22:      Update the total number of clusters $N_{cl}^{t+1} \leftarrow N_{cl}^t + \Delta N^t$
23: **end for**

---

**Theorem A.3.** *(Sufficient Condition for Sample Rate Conservation) Consider $K_{min}$ as the minimum number of clients permitted per cluster,* i.e. *the cardinality $|\mathcal{C}_n| \geq K_{min}$ for any given cluster $n = 1, \ldots, n_{cl}$, and $\rho \in (0, 1]$ to represent the initial sample rate. There exists a critical threshold $n^* > 0$ such that, if $K_{min} \geq n^*$ is met, the total sample size does not increase.*

*Proof.* Let us denote by $\rho_n$ the participation rate relative to the $n$-th cluster, *i.e.*

$$\rho_n = \max\left\{\rho, \frac{3}{|\mathcal{C}_n|}\right\} \tag{27}$$

because, in order to maintain privacy of the clients' information we need to sample at least three clients, therefore $\rho^n$ is at least 3 over the number of clients belonging to the cluster. The total participation rate at the end of the clustering process is given by

$$\rho^{\text{global}} = \sum_{n=1}^{n_{cl}} \frac{K_n}{K} \tag{28}$$

where $K_n$ denotes the number of clients sampled within the $n$-th cluster. If we focus on the term $K_n$, recalling Equation 27, we have that

$$K_n = \rho_n |\mathcal{C}_n| = \max\left\{\rho, \frac{3}{|\mathcal{C}_n|}\right\} \times |\mathcal{C}_n| = \max\{\rho|\mathcal{C}_n|, 3\} \ . \tag{29}$$

If we write Equation 29, by the means of the positive part function, denoted by $(x)^+ = \max\{0, x\}$, we obtain that

$$K_n = 3 + \max\{0, \rho|\mathcal{C}_n| - 3\} = 3 + (\rho|\mathcal{C}_n| - 3)^+ \ . \tag{30}$$

Observe that we are looking for a threshold value for which $\rho^{\text{global}} = \rho$, *i.e.* the participation rate remains the same during the whole training process.

Let us observe that $K_n = \rho|\mathcal{C}_n| \iff \rho|\mathcal{C}_n| \geq 3 \iff |\mathcal{C}_n| \geq n^* = 3/\rho$. In fact, if we assume that $K_{min} \geq n^*$, then the following chain of equalities holds

$$\rho^{\text{global}} = \sum_{n=1}^{n_{cl}} \frac{K_n}{K} = \frac{1}{K} \sum_{n=1}^{n_{cl}} \rho|\mathcal{C}_n| = \frac{\rho}{K} \sum_{n=1}^{n_{cl}} |\mathcal{C}_n| = \frac{\rho K}{K} = \rho$$

thus proving that $K_{min} \geq n^*$ is a sufficient condition for not increasing the sampling rate during the training process. $\square$

## B  THEORETICAL CONSTRUCTION OF THE CLUSTERING METRIC

To address the lack of clustering evaluation metrics suited for FL with distributional heterogeneity and class imbalance, we introduced a theoretically grounded adjustment to standard metrics, derived from the Wasserstein distance, Kantorovich–Rubinstein metric (Kantorovich, 1942). This metric, integrated with popular scores like Silhouette and Davies-Bouldin, enables a modular framework for a posteriori evaluation, effectively comparing clustering outcomes across federated algorithms. In this paragraph, we show how the proposed clustering metric that accounts for class imbalance can be derived from a probabilistic interpretation of clustering.

**Definition B.1.** *Let $(M, d)$ be a metric space, and $p \in [1, \infty]$. The Wasserstein distance between two probability measures $\mathbb{P}$ and $\mathbb{Q}$ over $M$ is defined as*

$$W_p(\mathbb{P}, \mathbb{Q}) = \inf_{\gamma \in \Gamma(\mathbb{P}, \mathbb{Q})} \mathbb{E}_{(x,y) \sim \gamma}[d(x, y)^p]^{1/p} \tag{31}$$

*where $\Gamma(\mathbb{P}, \mathbb{Q})$ is the set of all the possible couplings of $\mathbb{P}$ and $\mathbb{Q}$ (see Def. B.2).*

Furthermore, we need to introduce the notion of coupling of two probability measures.

**Definition B.2.** *Let $(M, d)$ be a metric space, and $\mathbb{P}, \mathbb{Q}$ two probability measures over $M$. A coupling $\gamma$ of $\mathbb{P}$ and $\mathbb{Q}$ is a joint probability measure on $M \times M$ such that, for any measurable subset $A \subset M$,*

$$\int_A \left(\int_M \gamma(dx, dy)\mathbb{Q}(dy)\right) \mathbb{P}(dx) = \mathbb{P}(A),$$
$$\int_A \left(\int_M \gamma(dx, dy)\mathbb{P}(dx)\right) \mathbb{Q}(dy) = \mathbb{Q}(A). \tag{32}$$

Let us recall that the empirical measure over $M$ of a sample of observations $\{x_1, \cdots, x_N\}$ is defined such that for any measurable set $A \subset M$

$$\mathbb{P}(A) = \frac{1}{N} \sum_{i=1}^{C} \delta_{x_i}(A) \tag{33}$$

where $\delta_{x_i}$ is the Dirac's measure concentrated on the data point $x_i$.

In particular, we aim to measure the goodness of a cluster by taking into account the distance between the empirical frequencies between two clients' class distributions and use that to properly

adjust the clustering metric. For the sake of simplicity, we assume that the distance $d$ over $M$ is the $L^2$-norm. We obtain the following theoretical result to justify the rationale behind our proposed metric.

**Theorem B.1.** *Let $s$ be an arbitrary clustering score. Then, the class-imbalance adjusted score $\tilde{s}$ is exactly the metric $s$ computed with the Wasserstein distance between the empirical measures over each client's class distribution.*

*Proof.* Let us consider two clients; each one has its own sample of observations $\{x_1, \ldots, x_C\}$ and $\{y_1, \ldots, y_C\}$ where the $i$-th position corresponds to the frequency of training points of class $i$ for each client. We aim to compute the $p$-Wasserstein distance between the empirical measures $\mathbb{P}$ and $\mathbb{Q}$ of the two clients, in particular for any $dx, dy > 0$

$$
\mathbb{P}(dx) = \frac{1}{N} \sum_{i=1}^{N} \delta_{x_i}(dx),
$$
$$
\mathbb{Q}(dy) = \frac{1}{N} \sum_{i=1}^{N} \delta_{y_i}(dy) \ .
\tag{34}
$$

In order to compute $W_p^p(\mathbb{P}, \mathbb{Q})$ we need to carefully investigate the set of all possible coupling measures $\Gamma(\mathbb{P}, \mathbb{Q})$. However, since either $\mathbb{P}$ and $\mathbb{Q}$ are concentrated over countable sets, it is possible to see that the only possible couplings satisfying Eq. 32 are the Dirac's measures over all the possible permutations of $x_i$ and $y_i$. In particular, by fixing the ordering of $x_i$, according to the rank statistic $x_{(i)}$, the coupling set can be written as

$$
\Gamma(\mathbb{P}, \mathbb{Q}) = \left\{ \frac{1}{C} \delta_{(x_{(i)}, y_{\pi(i)})} : \pi \in \mathcal{S} \right\}
\tag{35}
$$

where $\mathcal{S}$ is the set of all possible permutations of $C$ elements. Therefore we could write Eq. 31 as follows

$$
W_p^p = \min_{\pi \in \mathcal{S}} \int_{M \times M} |x - y|^p \frac{1}{N} \sum_{i=1}^{C} \delta_{(x_{(i)}, y_{\pi(i)})}(dx, dy)
\tag{36}
$$

since $\mathcal{S}$ is finite, the infimum is a minimum. By exploiting the definition of Dirac's distribution and the linearity of the Lebesgue integral, for any $\pi \in \mathcal{S}$, we get

$$
\int_{M \times M} |x - y|^p \frac{1}{C} \sum_{i=1}^{C} \delta_{(x_{(i)}, y_{\pi(i)})}(dx, dy) = \frac{1}{C} \sum_{i=1}^{C} \int_{M \times M} |x - y|^p \delta_{(x_{(i)}, y_{\pi(i)})}(dx, dy)
$$
$$
= \frac{1}{C} \sum_{i=1}^{C} |x_{(i)} - y_{\pi(i)}|^p \ .
\tag{37}
$$

Therefore, finding the Wasserstein distance between $\mathbb{P}$ and $\mathbb{Q}$ boils down to a combinatorial optimization problem, that is, finding the permutation $\pi \in \mathcal{S}$ that solves

$$
W_p^p(\mathbb{P}, \mathbb{Q}) = \min_{\pi \in \mathcal{S}} \frac{1}{C} \sum_{i=1}^{C} |x_{(i)} - y_{\pi(i)}|^p \ .
\tag{38}
$$

The minimum is achieved when $\pi = \pi^*$ that is the permutation providing the ranking statistic, i.e. $\pi^*(y_i) = y_{(i)}$, since the smallest value of the sum is given for the smallest fluctuations. Thus we conclude that the $p$-Wasserstein distance between $\mathbb{P}$ and $\mathbb{Q}$ is given by

$$
W_p(\mathbb{P}, \mathbb{Q}) = \left( \frac{1}{C} \sum_{i=1}^{C} |x_{(i)} - y_{(i)}|^p \right)^{1/p}
\tag{39}
$$

that is the pairwise distance computed between the class frequency vectors, sorted in order of magnitude, for each client, introduced in Section 3.5, where we chose $p = 2$. $\qquad\square$

## C  PRIVACY OF FEDGWC

In the framework of FedGWC, clients are required to send only the empirical loss vectors $l_k^{t,s}$ to the server (Cho et al., 2022). While concerns might arise regarding the potential leakage of sensitive information from sharing this data, it is important to clarify that the server only needs to access aggregated statistics, working on aggregated data. This ensures that client-specific information remains private. Privacy can be effectively preserved by implementing the Secure Aggregation protocol (Bonawitz et al., 2016), which guarantees that only the aggregated results are shared, preventing the exposure of any raw client data.

## D  COMMUNICATION AND COMPUTATIONAL OVERHEAD OF FEDGWC

FedGWC minimizes communication and computational overhead, aligning with the requirements of scalable FL systems (McMahan et al., 2016). On the client side, the computational cost remains unchanged compared to the chosen FL aggregation, e.g. FedA, as clients are only required to communicate their local models and a vector of empirical losses after each round. The size of this loss vector, denoted by $S$, corresponds to the number of local iterations (*i.e.* the product of local epochs and the number of batches) and is negligible w.r.t. the size of the model parameter space, $|\Theta|$. In our experimental setup, $S = 8$, ensuring that the additional communication overhead from transmitting loss values is negligible in comparison to the transmission of model weights.

All clustering computations, including those based on interaction matrices and Gaussian weighting, are performed exclusively on the server. This design ensures that client devices are not burdened with additional computational complexity or memory demands. The interaction matrix $P$ used in FedGWC is updated incrementally and involves sparse matrix operations, which significantly reduce both memory usage and computational costs.

These characteristics make FedGWC particularly well-suited for cross-device scenarios involving large federations and numerous communication rounds. Moreover, by operating on scalar loss values rather than high-dimensional model parameters, the clustering process in FedGWC achieves computational efficiency while maintaining effective grouping of clients. The server-side processing ensures that the method remains scalable, even as the number of clients and communication rounds increases. Consequently, FedGWC meets the fundamental objectives of FL by minimizing costs while preserving privacy and maintaining high performance.

## E  METRICS USED FOR EVALUATION

### E.1  SILHOUETTE SCORE

Silhouette Score is a clustering metric that measures the consistency of points within clusters by comparing intra-cluster and nearest-cluster distances (Rousseeuw, 1987). Let us consider a metric space $(M, d)$. For a set of points $\{x_1, \ldots, x_N\} \subset M$ and clustering labels $\mathcal{C}_1, \ldots, \mathcal{C}_{n_{cl}}$. The Silhouette score of a data point $x_i$ belonging to a cluster $\mathcal{C}_i$ is defined as

$$s_i = \frac{b_i - a_i}{\max\{a_i, b_i\}} \tag{40}$$

where the values $b_i$ and $a_i$ represent the average intra-cluster distance and the minimal average outer-cluster distance, *i.e.*

$$a_i = \frac{1}{|\mathcal{C}_i| - 1} \sum_{x_j \in \mathcal{C}_i \setminus \{x_i\}} d(x_i, x_j)$$
$$b_i = \min_{j \neq i} \frac{1}{|\mathcal{C}_j|} \sum_{x_j \in \mathcal{C}_j} d(x_i, x_j) \tag{41}$$

The value of the Silhouette score ranges between $-1$ and $+1$, *i.e.* $s_i \in [-1, 1]$. In particular, a Silhouette score close to 1 indicates well-clustered data points, 0 denotes points near cluster boundaries, and -1 suggests misclassified points. In order to evaluate the overall performance of the clustering, a common choice, that is the one adopted in this paper, is to average the score value for each data point.

### E.2 DAVIES-BOULDIN SCORE

The Davies-Bouldin Score is a clustering metric that evaluates the quality of clustering by measuring the ratio of intra-cluster dispersion to inter-cluster separation (Davies & Bouldin, 1979). Let us consider a metric space $(M, d)$, a set of points $\{x_1, \ldots, x_N\} \subset M$, and clustering labels $\mathcal{C}_1, \ldots, \mathcal{C}_{n_{cl}}$. The Davies-Bouldin score is defined as the average similarity measure $R_{ij}$ between each cluster $\mathcal{C}_i$ and its most similar cluster $\mathcal{C}_j$:

$$DB = \frac{1}{n_{cl}} \sum_{i=1}^{n_{cl}} \max_{j \neq i} R_{ij} \tag{42}$$

where $R_{ij}$ is given by the ratio of intra-cluster distance $S_i$ to inter-cluster distance $D_{ij}$, *i.e.*

$$R_{ij} = \frac{S_i + S_j}{D_{ij}} \tag{43}$$

with intra-cluster distance $S_i$ defined as

$$S_i = \frac{1}{|\mathcal{C}_i|} \sum_{x_k \in \mathcal{C}_i} d(x_k, c_i) \tag{44}$$

where $c_i$ denotes the centroid of cluster $\mathcal{C}_i$, and $D_{ij} = d(c_i, c_j)$ is the distance between centroids of clusters $\mathcal{C}_i$ and $\mathcal{C}_j$. A lower Davies-Bouldin Index indicates better clustering, as it reflects well-separated and compact clusters. Conversely, a higher DBI suggests that clusters are less distinct and more dispersed.

#### E.2.1 RAND INDEX

Rand Index is a clustering score that measures the outcome of a clustering algorithm with respect to a ground truth clustering label (Rand, 1971). Let us denote by $a$ the number of pairs that have been grouped in the same clusters, while by $b$ the number of pairs that have been grouped in different clusters, then the Rand-Index is defined as

$$RI = \frac{a + b}{\binom{N}{2}} \tag{45}$$

where N denotes the number of data points. In our experiments we opted for the Rand Index score to evaluate how the algorithm was able to separate clients in groups of the same level of heterogeneity (which was known a priori and used as ground truth). A Rand Index ranges in $[0, 1]$, and a value of 1 signifies a perfect agreement between the identified clusters and the ground truth.

## F ESTIMATING THE DIRICHLET PARAMETER $\alpha$

In this appendix, we provide a detailed explanation of the procedure used to estimate the Dirichlet parameter $\beta$ for each split identified by our algorithm. This analysis is conducted a posteriori to statistically evaluate the capability of `FedGWC` in grouping clients with similar levels of data heterogeneity.

Consider a fixed cluster of clients, denoted by $\mathcal{C}_i$. For each client $k \in \mathcal{C}_i$, let $z_k \in \mathbb{N}^C$ represent the vector of sample counts per class (with $C$ as the total number of classes). Given $z_k$, we can compute the likelihood function, assuming $z_k$ is drawn from a Dirichlet distribution parameterized by the skew parameter $\alpha_k$, such that $\mathcal{L}(\alpha_k; z_k)$ represents the likelihood function for $z_k$.

To estimate $\alpha_k$, we employ the maximum likelihood estimator by solving:

$$\hat{\alpha}_k \in \arg \max_{\alpha > 0} \mathcal{L}(\alpha; z_k)$$

This estimation is achieved through stochastic optimization (e.g., ADAM or SGD). For each cluster, we obtain an estimate $\hat{\alpha}_k$, which allows us to compare average values, standard errors, and confidence intervals of $\alpha$ across clusters detected by `FedGWC`, providing a quantitative measure of heterogeneity among clusters.

## G    Datasets and implementation details

To simulate a realistic FL environment with heterogeneous data distribution, we conduct experiments on CIFAR-100 (Krizhevsky et al., 2009). As a comparison, we also run experiments on the simpler CIFAR-10 dataset Krizhevsky et al. (2009). CIFAR-10 and CIFAR-100 are distributed among $K$ clients using a Dirichlet distribution (by default, we use $\alpha = 0.05$ for CIFAR-10 and $\alpha = 0.5$ for CIFAR-100) to create highly imbalanced and heterogeneous settings. By default, we use $K = 100$ clients with 500 training and 100 test images. The classification model is a CNN with two convolutional blocks and three dense layers. Additionally, we perform experiments on the Femnist dataset LeCun (1998), partitioned among 400 clients using a Dirichlet distribution with $\alpha = 0.01$. In these experiments, we employ LeNet5 as the classification model LeCun et al. (1998). Local training on each client uses SGD with a learning rate of 0.01, weight decay of $4 \cdot 10^{-4}$, and batch size 64. The number of local epochs is 1, resulting in 7 batch iterations for CIFAR-10 and CIFAR-100 and 8 batch iterations for Femnist. The number of communication rounds is set to 3,000 for Femnist, 10,000 for CIFAR-10 and 20,000 for CIFAR-100, with a 10% client participation rate per cluster. For FedGWC we employ constant value $\alpha_t = \alpha$ equal to the participation rate, *i.e.* 10%. As the performance metric, we use the balanced accuracy, *i.e.* the average accuracy of each cluster model evaluated on the test sets associated to each client in the same cluster.

## H    Sensitive Analysis beta value RBF kernel

This section provides a sensitivity analysis for the $\beta$ hyper-parameter of the RBF kernel adopted for FedGWC. The results of this tuning are shown in Table 6.

Table 6: A sensitivity analysis on the RBF kernel hyper-parameter $\beta$ is conducted. We present the balanced accuracy for FedGWC on the Cifar10, Cifar100, and Femnist datasets for $\beta \in \{0.1, 0.5, 1.0, 2.0, 4.0\}$. It is noteworthy that FedGWC demonstrates robustness to variations in this hyperparameter.

| $\beta$ | Cifar10 | Cifar100 | Femnist |
|---|---|---|---|
| 0.1 | 74.2 | 49.9 | 76.0 |
| 0.5 | 74.9 | **53.4** | 76.0 |
| 1.0 | 75.1 | 49.5 | 76.0 |
| 2.0 | **75.6** | 50.9 | 75.6 |
| 4.0 | 72.6 | 52.6 | **76.1** |

## I    Evaluation of IFCA and CFL algorithms with different number of clusters

This section shows the tuning of the number of clusters for the IFCA and CFL algorithms, which cannot automatically detect this value. The results of this tuning are shown in Table 7.

## J    Further Experiments

Figure 5 illustrates the clustering results corresponding to varying degrees of heterogeneity, as described in Section 4.2. As per FedGWC, the detection of clusters based on different levels of heterogeneity in the Cifar10 dataset is achieved. Specifically, an examination of the interaction matrix reveals a clear distinction between the two groups. In Figure 6, we show that in class-balanced scenarios with small heterogeneity, like Cifar10 with $\alpha = 100$, FedGWC successfully detects one single cluster. Indeed, in homogeneous scenarios such as this one, the model benefits from accessing more data from all the clients.

Figure 4 shows how the MSE converges to a small value as the rounds increase for a Cifar10 experiment.

As Figure 7 illustrates, FedGWC partitions the CIFAR-100 dataset into clients based on class distributions. Each cluster's distribution is distinct and non-overlapping, demonstrating the algorithm's efficacy in partitioning data with varying degrees of heterogeneity. In Figure 8, we report the domain detection on Cifar100, where 40 clients have clean images, 30 have noisy images, and 30 have blurred images. Table 5 shows that FedGWC performs a good clustering, effectively separating the different domains.

Table 7: Performance of for baseline algorithms for clustering in FL `FeSEM`, and `IFCA`, w.r.t. the number of clusters

| | | Clustering method | C | Acc |
|---|---|---|---|---|
| **Dataset** | Cifar10 | IFCA | 2 | $78.6 \pm 2.3$ |
| | | | 3 | $75.7 \pm 5.2$ |
| | | | 4 | $71.9 \pm 10.2$ |
| | | | 5 | $78.4 \pm 2.3$ |
| | | FeSem | 2 | $70.0 \pm 4.2$ |
| | | | 3 | $67.5 \pm 1.3$ |
| | | | 4 | $70.5 \pm 1.6$ |
| | | | 5 | $68.7 \pm 0.8$ |
| | Cifar100 | IFCA | 2 | $46.7 \pm 0.0$ |
| | | | 3 | $44.0 \pm 1.6$ |
| | | | 4 | $45.1 \pm 2.6$ |
| | | | 5 | $47.5 \pm 3.5$ |
| | | FeSem | 2 | $43.3 \pm 1.3$ |
| | | | 3 | $48.0 \pm 1.9$ |
| | | | 4 | $50.9 \pm 1.8$ |
| | | | 5 | $53.4 \pm 1.8$ |
| | Femmist | IFCA | 2 | $76.1 \pm 0.1$ |
| | | | 3 | $75.9 \pm 1.9$ |
| | | | 4 | $76.6 \pm 0.1$ |
| | | | 5 | $76.7 \pm 0.6$ |
| | | FeSem | 2 | $75.6 \pm 0.2$ |
| | | | 3 | $75.5 \pm 0.5$ |
| | | | 4 | $75.0 \pm 0.1$ |
| | | | 5 | $74.9 \pm 0.1$ |

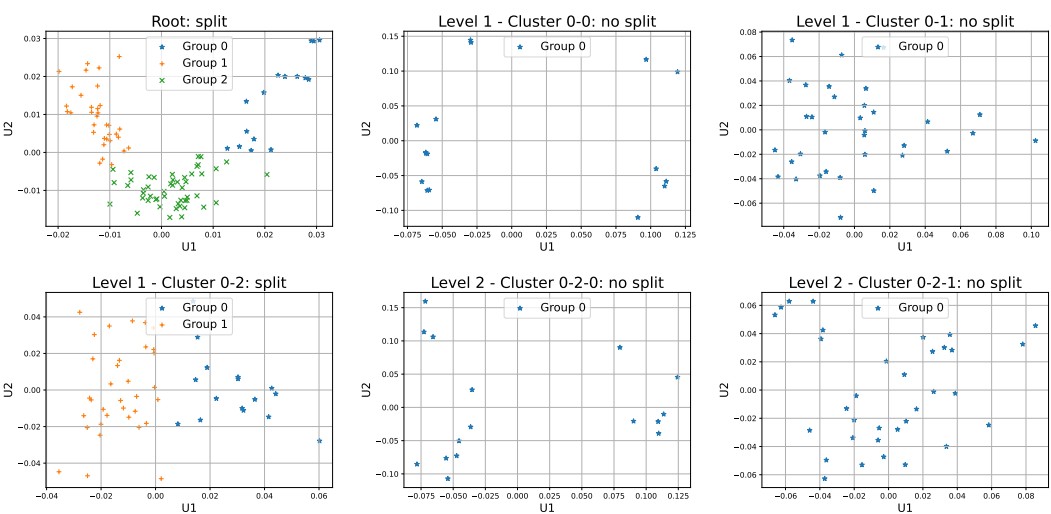

Figure 3: Cluster evolution with respect to the recursive splits in `FedGWC` on Cifar100, projected on the spectral embedded bi-dimensional space. From left to right, top to bottom, we can see that `FedGWC` splits the client into cluster, until a certain level of intra-cluster homogeneity is reached

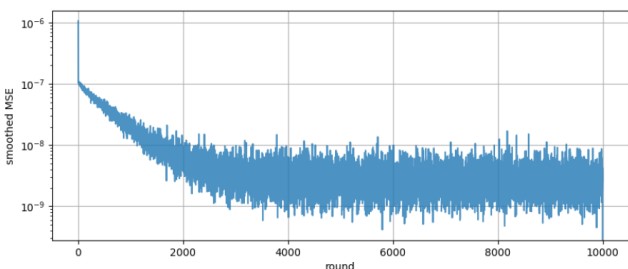

Figure 4: Interaction matrix convergence: on the $y$-axis MSE, computed as in Eq. 7 in logarithmic scale w.r.t. communication rounds in the $x$-axis on CIFAR-10, with Dirichlet parameter $\alpha = 0.05$.

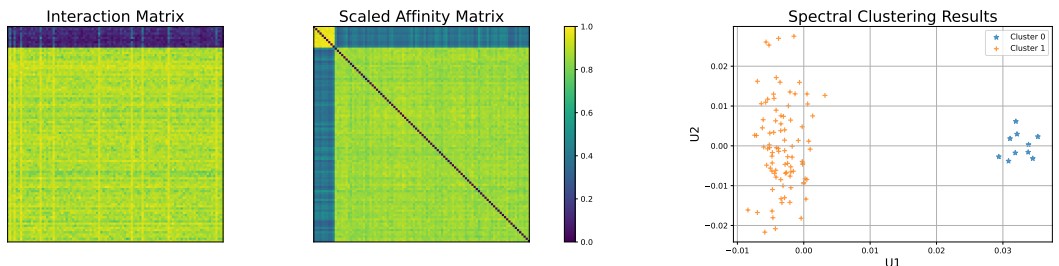

Figure 5: Homogeneous (CIFAR-10 $\alpha = 100$) vs heterogeneous clustering (CIFAR-10 $\alpha = 0.05$). The interaction matrix at convergence and the corresponding scaled affinity matrix are on the left. The scatter plot in the 2D plane with spectral embedding is on the right. It is possible to see that the algorithm perfectly separates homogeneous clients (orange) from heterogeneous clients (blue)

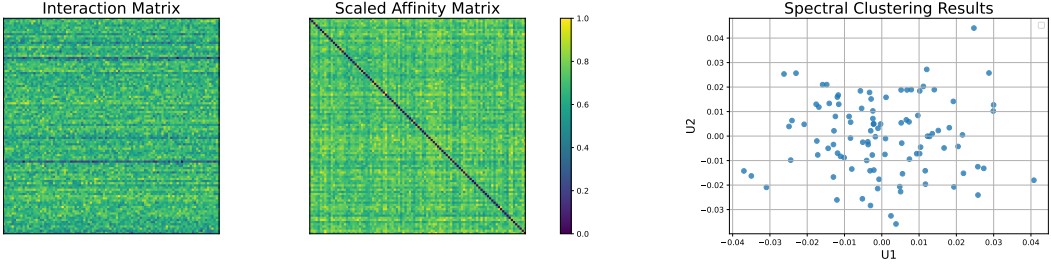

Figure 6: Homogeneous case (CIFAR-10 $\alpha = 100$). The interaction matrix at convergence and the corresponding scaled affinity matrix are on the left. The scatter plot in the 2D plane with spectral embedding is on the right. In the homogeneous case where no clustering is needed, *FedGW* does not split the clients.

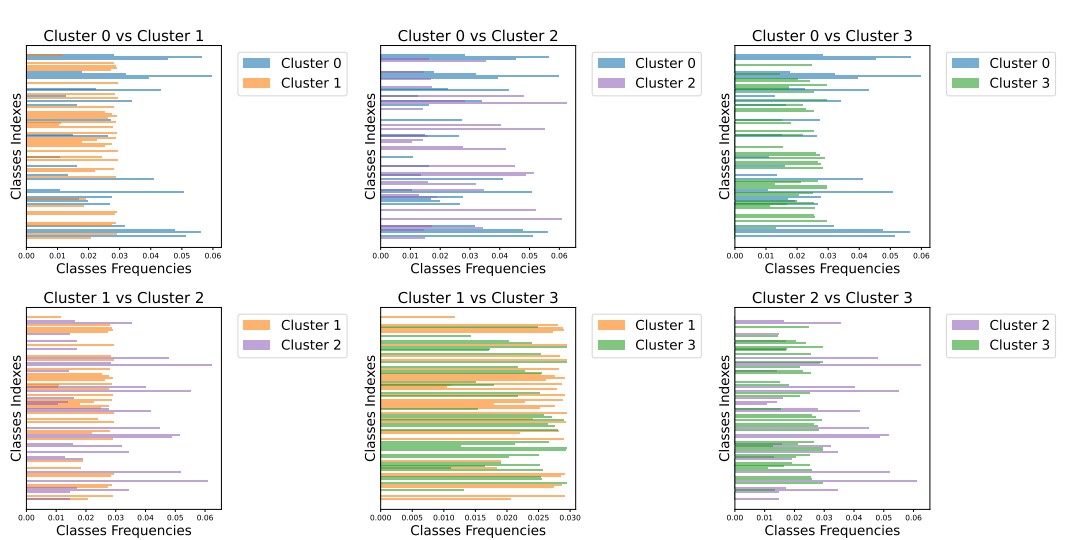

Figure 7: Class distributions among distinct clusters as detected by FedGWC on Cifar100. Specifically, we examine the class distributions for each pair of clusters, demonstrating that (1) the clusters were identified by grouping differing levels of heterogeneity and (2) there is, in most cases, an absence of overlapping classes.

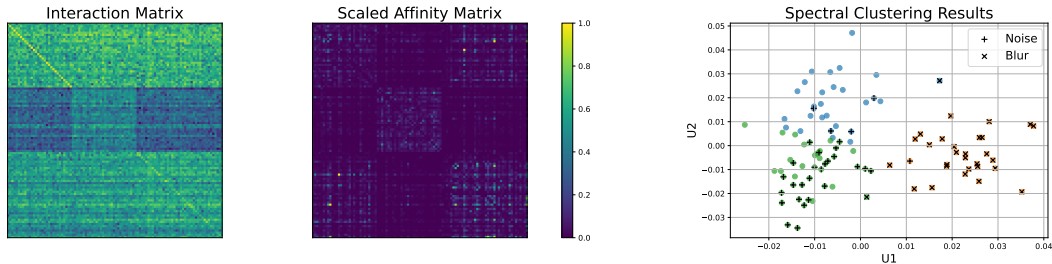

Figure 8: FedGWC in the presence of domain imbalance. Three domains on Cifar100: clean clients (unlabeled), noisy clients (+), and blurred clients (x). *Left*: is the interaction matrix $P$ at convergence from which it is possible to see client relations. *Center*: The affinity matrix $W$ computed with respect to the UPVs extracted from $P$, and on which FedGW_Clustering is performed. We can see that FedGWC clusters the clients according to the domain, as proved by results in Table 5.

