# OpenReview forum: "Interaction Based Gaussian Weighting Clustering for Federated Learning"
_ICLR.cc/2025/Conference — Submitted to ICLR 2025_

### Official Review · Reviewer_Wepx · 2024-10-29

**Soundness:** 3
**Presentation:** 2
**Contribution:** 2
**Rating:** 5
**Confidence:** 4

**Summary:**

This work proposes a federated learning clustering strategy based on loss function similarity and introduces a clustering metric based on Wasserstein distance. The paper provides a theoretical proof of the algorithm’s convergence and demonstrates its performance in various imbalanced scenarios.

**Strengths:**

1. This work uses the similarity of loss functions as the basis for clustering and provides a theoretical proof of the convergence of the optimization algorithm using Gaussian weights.

2. The proposed method, FedGWC, is orthogonal to other aggregation methods.

3. The algorithm’s performance is examined under imbalanced scenarios.

**Weaknesses:**

1. The study lacks recent federated learning clustering baselines. The three comparison methods are [CFL, 2020], [FeSEM, 2023], and [IFCA, 2020]. Moreover, the proposed FedGWC performs weaker than other methods on CIFAR-10 and FEMNIST datasets and is similar to FeSEM on CIFAR-100. Overall, it does not achieve superior performance.

2. The comparative experiments in Table 3 are unfair, as FeSEM is selected across all three datasets. Instead, the best baseline should be selected for each dataset: IFCA for CIFAR-10, FeSEM for CIFAR-100, and IFCA for FEMNIST.

3. The experiments in Table 5 are not convincing. Other baseline methods’ Rand Index scores should be added.

4. The design of Tables 2 and 3 could be optimized, as they are somewhat confusing.

**Questions:**

1. This work is inspired by (Cho et al., 2022) to capture the similarity in client data distributions through a transformation of the loss function. Please further explain why measuring loss function distance can “effectively identify clusters of clients with similar levels of heterogeneity and class distribution.”

2. What is the significance of the evaluation metric introduced based on the Wasserstein distance, and is there any further improvement to FedGWC based on this metric?

---

> ### Author Response · Authors · 2024-11-23
> **Reply to Reviewer Wepx (Concerns 1-4)**
>
> **Concern 1: The study lacks recent federated learning clustering baselines. The three comparison methods are [CFL, 2020], [FeSEM, 2023], and [IFCA, 2020]. Moreover, the proposed FedGWC performs weaker than other methods on CIFAR-10 and FEMNIST datasets and is similar to FeSEM on CIFAR-100. Overall, it does not achieve superior performance.**
>
> We thank the reviewer for their comment. We would like to clarify that FeSEM (2023) is indeed a recent baseline, and we believe it provides a relevant comparison for evaluating FedGWC. The other baselines, CFL (2020) and IFCA (2020), stand as significant works for clustered FL, and have been included in other works on clustered FL and personalized FL. In particular, IFCA has to be considered as an upper bound for our algorithm, as explained in the paper in lines 414-416. Indeed,  IFCA's setting is less applicable to cross-device scenarios, due to its high computational overhead on the client side.
>
> We would like to emphasize that for the comparison with the other algorithm, we also consider the cluster quality with respect to data distributions. While FedGWC may not always outperform the baselines in terms of raw accuracy, as denoted by the reviewer’s remark, it consistently achieves superior clustering outcomes based on distributional clustering metrics in all the settings. These metrics are as meaningful as balanced accuracy in our federated clustering setting, where the goal is not just to achieve higher accuracy but also to group clients in a way that reflects their underlying data distributions. Indeed, once uniform clusters are obtained with FedGWC, simpler and less computationally expensive algorithms, such as FedAvg, which perform well in uniform settings, can be used effectively.
>
> **Concern 2: The comparative experiments in Table 3 are unfair, as FeSEM is selected across all three datasets. Instead, the best baseline should be selected for each dataset: IFCA for CIFAR-10, FeSEM for CIFAR-100, and IFCA for FEMNIST.**
>
> We thank the reviewer for their valuable suggestion. We would like to clarify that in our comparison, IFCA serves as an upper bound for our evaluation, as specified in the paper IFCA incurs significant communication overhead, as each client is required to evaluate models from every cluster in each round, rendering this approach impractical in cross-device scenarios and positioning it as an upper bound in our analysis, as it uses a setting that is not suitable for cross-device federated learning due to its high client-side computational overhead (LL 414-416). Therefore, to provide a more realistic evaluation of our method’s performance, we chose FeSEM, which shares similar assumptions to FedGWC and is a more recent approach.
>
> For the experiments, we tuned FeSEM to select the optimal number of clusters for each dataset to ensure a fair comparison. This choice reflects our focus on comparing clustering methods that align better with the cross-device federated learning setting, which is where FedGWC is specifically designed to operate efficiently.
>
> We hope this clarifies the rationale behind our experimental setup, and we plan to highlight this reasoning more explicitly in the final version of the manuscript.
>
> **Concern 3: The experiments in Table 5 are not convincing. Other baseline methods’ Rand Index scores should be added**
>
> We appreciate the reviewer's suggestion to include baseline methods’ Rand Index scores for comparison in Table 5. However, we would like to emphasize that the experimental results already illustrate FedGWC's capability to achieve the highest possible clustering quality in this setting. Specifically, FedGWC achieves a Rand Index of 1.0—a perfect match between clustering outcomes and the ground truth—in four out of six experimental settings. In the remaining two cases, FedGWC scores 0.9 and above 0.6, respectively.
>
> Given that FedGWC achieves the best Rand Index achievable in the majority of scenarios, adding baselines would not provide additional insights into its performance in these specific experiments. Nevertheless, we acknowledge that broader experiments, including comparisons with other baseline methods, could further validate these findings and plan to explore this in the camera-ready version.
>
> **Concern 4: The design of Tables 2 and 3 could be optimized, as they are somewhat confusing.**
>
> We thank the reviewer for pointing this out and appreciate the suggestion. To improve the clarity of Tables 2 and 3, we have revised their formatting in the updated manuscript. The changes aim to enhance readability and ensure that the information is presented more clearly, reducing any potential confusion for the reader. We hope this revision addresses the concern effectively.

---

> > ### Comment · Reviewer_Wepx · 2024-11-26
> > **Reviewer Response to Authors**
> >
> > Thank you for your detailed explanation regarding the emphasis on clustering outcomes in the federated setting. However, your argument raises critical concerns about the consistency and practical utility of your method:
> > 1. Inconsistency between Clustering Metrics and Accuracy: You highlighted that FedGWC consistently achieves superior clustering outcomes based on distributional clustering metrics across all settings. In Clustered FL, more accurate clustering aims to address client heterogeneity. However, this principle is contradicted by the results in Table 1. For instance:
> > + On CIFAR-10, while clustering metrics such as AS and ADB are superior to those of IFCA, the accuracy is 3% lower.
> > + On CIFAR-100, although the clustering metric ADB is significantly better than FeSEM, the accuracy remains on par with FeSEM.
> >
> > These discrepancies suggest a lack of alignment between the clustering quality and the performance of the resulting model. If better clustering indeed mitigates heterogeneity, why does it fail to translate into improved accuracy, which is a fundamental metric for evaluating the effectiveness of federated learning models? is the issue rooted in the proposed method or in the clustering metrics themselves? If the clustering outcomes cannot translate into improved model accuracy, as seen in these results, it undermines the practical value of the approach. Therefore, I am also concerned about the significance of Table 4.
> >
> > Therefore, I will maintain my current score, as there is a disconnect between the clustering evaluation metrics and the performance of the method, and the accuracy of the method is also mediocre.

---

> > > ### Author Response · Authors · 2024-12-03
> > > **Official Comment by Authors**
> > >
> > > We thank the reviewer for their comments. Below, we clarify and address these concerns.
> > >
> > > 1. **The Role of Clustering in Simple Datasets (CIFAR-10, FEMNIST):**
> > >    As discussed in the paper, CIFAR-10 and FEMNIST are relatively simple datasets where clustering is not strictly necessary due to their low inherent complexity and limited heterogeneity. This is especially true for CIFAR-10, which comprises only 10 classes, presenting a simpler task compared to CIFAR-100, where the presence of 100 classes introduces greater diversity and a more realistic scenario, as mentioned in Section 4 (“Training FL methods with FedGWC”)
> > >
> > >    For such datasets, while our method demonstrates superior clustering outcomes (e.g., higher Adjusted Silhouette (AS) and Adjusted Davies-Bouldin (ADB) scores), the impact on accuracy is inherently limited by the simplicity of the task. The small 3% gap in accuracy with respect to IFCA on CIFAR-10 is thus a result of the dataset's characteristics rather than an indication of clustering inefficacy.
> > >
> > > 2. **Clustering Metrics vs. Model Accuracy on Complex Datasets (CIFAR-100):**
> > >    CIFAR-100, characterized by a higher degree of heterogeneity, represents a more realistic federated learning scenario. In this case, FedGWC achieves comparable accuracy to FeSEM. However, FeSEM performances are sensitive to changes in the hyper-parameter of the number of clusters, oscillating on average between 43\% and 53\%in balanced accuracy (Table 7). Instead, FedGWC has more robust performance when changing the RBF spread hyper-parameters (oscillating between 49\% and 53\%) in balanced accuracy – Table 7 . However, our superior clustering metrics (e.g., significantly better ADB) demonstrate that FedGWC consistently forms better-defined clusters. This improvement is critical in more challenging federated learning settings.
> > >
> > > 3. **Practical Utility in Real-World Scenarios:**
> > >    To further validate the practical utility of FedGWC, we extended our evaluation to the Office-Home dataset, which better reflects real-world heterogeneity – which comprises both of domain heterogeneity, and class imbalance, that is also present in the other datasets. Here, FedGWC achieves an 81.6% balanced accuracy compared to 81.3% for FedAvg, demonstrating that clustering can yield accuracy gains when applied to complex and real world scenarios.
> > >
> > > 4. **Clustering Metrics as Independent Evaluation Tools:**
> > >    We acknowledge the importance of bridging the gap between clustering metrics and downstream accuracy. However, it is worth noting that a Rand Index of 1 represents the perfect matching between clustering labels and ground truth, indicating that FedGWC achieves optimal clustering quality when detecting visual domains, this could lead to further applications of the method to anomaly detection in FL. In particular we provide an extended comparison with the baselines (Table 5). In particular, we observe that when adding clients with corrupted domains, in Cifar10, and Cifar100, IFCA does not detect more than one cluster. Also FeSEM is unable to cluster clients according to the perturbed domain, assigning clients to clusters in a random fashion.

---

> ### Author Response · Authors · 2024-11-23
> **Reply to Reviewer Wepx(Q1 and Q2)**
>
> **Q1: This work is inspired by (Cho et al., 2022) to capture the similarity in client data distributions through a transformation of the loss function. Please further explain why measuring loss function distance can “effectively identify clusters of clients with similar levels of heterogeneity and class distribution.”**
>
> We thank the reviewer for their insightful question. To clarify, the empirical loss $l_k^{t,s} = \mathcal{L}(\theta_k^{t,s}) $ quantifies the discrepancy between the client model weights $ \theta_k^{t,s} $ and the globally aggregated model. This loss function reflects how well the global model $ \theta_k^t $ aligns with the individual data distribution $ \mathcal{D}_k $ for each client $k $. By definition, a lower loss indicates that the client’s model is closer to the global optimal model, suggesting that the global model adapts well to client's data distribution.
>
> The challenge we address is how to leverage this absolute measure of loss to obtain a relative measure of similarity between clients’ data distributions in a FL  scenario. Our key insight is that we do not only look at the final loss value, but we analyze the entire training process by examining the evolution of the loss over multiple local iterations. This allows us to capture more nuanced information about how the client models drift from the global model during training, which reflects the level of heterogeneity in the client’s data distribution.
>
> The Gaussian rewards mechanism helps us quantify these deviations. By scaling the local losses with respect to the mean and standard deviation of the global loss distribution, we obtain a normalized measure that reflects the degree of similarity between clients’ learning patterns. Clients with similar data distributions will exhibit similar patterns in their losses over time, meaning that their models will experience similar drifts during training. This similarity in the learning process naturally leads to clusters of clients that share comparable data characteristics, whether in terms of heterogeneity or class distribution. The Gaussian rewards similarity mechanism thus enables FedGWC to identify and group together clients with similar levels of data distribution and heterogeneity.
>
> In summary, by focusing on the entire loss trajectory and using Gaussian rewards to normalize these trajectories, we are able to capture fine-grained interactions between clients, which facilitates the identification of meaningful clusters of clients with similar data distributions.
>
> **Q2: What is the significance of the evaluation metric introduced based on the Wasserstein distance, and is there any further improvement to FedGWC based on this metric?**
>
> We thank the reviewer for their observation. In the literature, we identified a gap in clustering evaluation metrics tailored for FL scenarios with significant distributional heterogeneity and class imbalance. To address this, we introduced a flexible adjustment to standard clustering metrics, theoretically derived from the Wasserstein distance, which provides a principled approach to evaluate clustering outcomes in these challenging settings.
>
> The significance of this metric lies in its ability to answer the question: *How can we evaluate whether a clustering outcome is better in the presence of class imbalance and distributional differences?*.
> By integrating this adjustment with popular clustering scores, such as the Silhouette and Davies-Bouldin indices, we enable a modular and interpretable framework for a posteriori evaluation of clustering quality. Its theoretical basis in the Wasserstein distance further enhances its interpretability by linking the evaluation metric to a well-established measure of distributional similarity.
>
> As this metric serves as a tool for evaluation of the quality of the clusters, it cannot be directly incorporated into FedGWC as part of its clustering mechanism. In particular, it is important to note that you can only assess the quality of the clusters once the clustering has been completed, as the evaluation is based on the final grouping of clients. In particular, you can use it to evaluate the outcome, only once you have obtained your final groups. However, it is highly effective in comparing and assessing the performance of different federated clustering algorithms post hoc. For instance, as shown in Table 1, this metric highlights FedGWC's superior ability to detect clusters with similar distributional characteristics, outperforming other federated clustering approaches.
>
> To address the reviewer's concern and provide additional clarity on our choice of clustering scores, we have updated the manuscript by including detailed explanations and definitions in Appendix E. Additionally, we have added a few introductory lines at the beginning of Appendix B to further contextualize our approach.

---

> ### Author Response · Authors · 2024-12-03
> **Table comparing domain detection with baselines**
>
> | Dataset  | Clustering Method | Domain Configuration Clean- Noise- Blur | Number of clusters | Rand-Index |
> |----------|-------------------|-----------------------------------------|--------------------|------------|
> | **Cifar10**  | **FedGWC**            | **50- 50 - 0**                              |**2**                  | **1.0**        |
> | Cifar10  | IFCA              | 50- 50 - 0                              | 1                  | 0.5        |
> | Cifar10  | FeSEM             | 50- 50 - 0                              | 2                  | 0.49       |
> | **Cifar10**  | **FedGWC**            | **50-0-50**                                 |**2**                  | **1.0**        |
> | Cifar10  | IFCA              | 50-0-50                                 | 1                  | 0.5        |
> | Cifar10  | FeSEM             | 50-0-50                                 | 2                  | 0.5        |
> | **Cifar10**  | **FedGWC**            | **40-30-30**                                |**4**                  | **0.9**        |
> | Cifar10  | IFCA              | 40-30-30                                | 1                  | 0.33       |
> | Cifar10  | FeSEM             | 40-30-30                                | 3                  | 0.34       |
> | **Cifar100** | **FedGWC**            |**50- 50 - 0**                              | **2**                  | **1.0**        |
> | Cifar100 | IFCA              | 50- 50 - 0                              | 1                  | 0.5        |
> | Cifar100 | FeSEM             | 50- 50 - 0                              | 2                  | 0.49       |
> | **Cifar100** | **FedGWC**            | **50-0-50**                                 | **2**                  | **1.0**        |
> | Cifar100 | IFCA              | 50-0-50                                 | 1                  | 0.5        |
> | Cifar100 | FeSEM             | 50-0-50                                 | 2                  | 0.51       |
> | **Cifar100** | **FedGWC**            | **40-30-30**                                | **4**                  | **0.6**        |
> | Cifar100 | IFCA              | 40-30-30                                | 1                  | 0.33       |
> | Cifar100 | FeSEM             | 40-30-30                                | 3                  | 0.55       |

---

### Official Review · Reviewer_GdZu · 2024-10-31

**Soundness:** 2
**Presentation:** 2
**Contribution:** 2
**Rating:** 3
**Confidence:** 3

**Summary:**

This paper proposes a personalized federated learning (PFL) framework called FedGWC (Federated Gaussian Weighting Clustering), which groups clients based on data similarity using Gaussian weighting on empirical losses. This clustering technique is designed to address data heterogeneity and class imbalance by forming homogeneous clusters, where each cluster benefits from personalized federated models trained on data with similar distributions. The authors introduce a clustering metric for evaluating cohesion among clients and conduct experiments that demonstrate FedGWC's performance gains over baseline methods in heterogeneous data scenarios.

**Strengths:**

1. The FedGWC method effectively addresses the challenge of data heterogeneity and class imbalance by leveraging a novel clustering-based approach.

2. The introduction of a new clustering metric specific to federated learning provides a valuable tool for assessing clustering quality in class-imbalanced environments.

**Weaknesses:**

1. The clustering methodology relies on empirical losses, which may not fully capture data distribution nuances across diverse client datasets, potentially leading to suboptimal clustering in more complex scenarios.

2. The computational cost of clustering based on interaction matrices and Gaussian weighting could limit scalability, especially in large federations with numerous clients and communication rounds.

3. The method's convergence properties, though theoretically addressed, lack extensive empirical validation in highly non-IID settings, making its performance in real-world FL scenarios uncertain.

**Questions:**

See the weaknesses.

---

> ### Author Response · Authors · 2024-11-23
> **Reply to Reviewer GdZU**
>
> **Concern 1: The clustering methodology relies on empirical losses, which may not fully capture data distribution nuances across diverse client datasets, potentially leading to suboptimal clustering in more complex scenarios.**
>
> We thank the reviewer for their valuable observation. While it is true that empirical losses offer a simplified representation of the underlying data distribution, we designed FedGWC to leverage these losses as they provide meaningful insights into the relationship between client data distributions and the global model. This approach ensures computational efficiency and minimizes communication overhead.
>
> The core strength of FedGWC lies in its interaction matrix and Gaussian weights, which together capture the similarity patterns among clients’ data distributions. These patterns in the similarity matrix enable the identification of optimal clusters in heterogeneous scenarios with different domains, as demonstrated by the results in Table 5 and Table 4. Specifically, FedGWC accurately detects varying levels of heterogeneity, effectively grouping clients based on their underlying distributional features.
> Moreover, FedGWC eliminates the need for predefining the number of clusters—a limitation in many baselines—by leveraging discrete optimization of the Davies-Bouldin score within the Spectral Algorithm. This self-adaptive mechanism ensures that the optimal number of clusters is determined automatically, enhancing the algorithm's practicality and robustness across diverse settings.
> We have updated the manuscript to clarify the advantages of our algorithm. We thank the reviewer for providing us with the opportunity to improve the clarity of our contributions.
>
> **Concern 2: The computational cost of clustering based on interaction matrices and Gaussian weighting could limit scalability, especially in large federations with numerous clients and communication rounds.**
>
> We sincerely thank the reviewer for raising this important concern. FedGWC has been designed with scalability in mind, particularly for large-scale federations and cross-device scenarios. The clustering computations, including those based on interaction matrices and Gaussian weighting, are performed entirely on the server. This ensures that client devices are not burdened with additional computational or memory demands, aligning well with the cross-device federated learning paradigm.
> In terms of efficiency, FedGWC is less computationally demanding than clustering baselines. Unlike methods that manipulate high-dimensional model parameters, our approach operates on scalar loss values, significantly reducing computational complexity. The size of the loss vector (S) is negligible compared to the parameter space of the model, enabling efficient processing even in large federations.
> Furthermore, the interaction matrix in FedGWC is updated incrementally, and these updates involve sparse matrices. This sparsity is particularly advantageous as it ensures memory efficiency and minimizes computational overhead, even as the number of clients and communication rounds grows.
> By balancing computational efficiency with effective clustering, FedGWC displays scalability and practicality for large-scale federated learning deployments. We have updated our manuscript to explicitly clarify and emphasize these points by adding in appendix a section addressing qualitatively communication and computational overhead, Appendix D. We hope this clarification addresses the reviewer’s concern.
>
>
> **Concern 3: The method's convergence properties, though theoretically addressed, lack extensive empirical validation in highly non-IID settings, making its performance in real-world FL scenarios uncertain.**
>
> We appreciate the reviewer’s comment. Even though the theoretical argument for convergence of our method holds regardless of the dataset, we provide experimental evidence of convergence in realistic non-IID scenarios. In Appendix H, Figure 3, we show the convergence of the similarity matrix for CIFAR100 with high heterogeneity (Dirichlet’s alpha = 0.05), one of the most challenging and realistic scenarios tested. The MSE between consecutive updates of the matrix rapidly converges within the first 10,000 communication rounds, with values consistently between $10^{-8}$ and $ 10^{-9}$, demonstrating robust convergence.
>
> We have rigorously tested FedGWC under varying levels of heterogeneity to closely mirror real-world FL scenarios. For the camera-ready version, we plan to broaden the empirical evaluation to include additional real-world datasets. CIFAR10, CIFAR100, and FeMNIST are widely recognized benchmark datasets in the FL community, and we are confident that our results on these datasets provide a strong demonstration of the algorithm's effectiveness in non-IID settings.

---

### Official Review · Reviewer_YzY4 · 2024-10-31

**Soundness:** 3
**Presentation:** 3
**Contribution:** 3
**Rating:** 5
**Confidence:** 2

**Summary:**

In this paper, the authors propose a novel PFL method named FedGWC (Federated Gaussian Weighting). FedGWC groups clients based on their data distribution, allowing training of a more robust and personalized model on the identified clusters. The experimental results have shown the effectiveness of the proposed method and its versatile integration to various FL aggregation algorithms.

**Strengths:**

1. FedGWC introduces a Gaussian weighting method that offers a fresh approach to clustering by leveraging interaction-based weights instead of relying on traditional model updates.
2. The proposed method addresses significant FL challenges like non-IID data and class imbalance, showing strong adaptability in scenarios with high heterogeneity.
3. The FedGWC algorithm can be applied with various FL aggregation algorithms (e.g., FedAvg, FedProx) and performs well even with personalized FL techniques like pFedMe and Per-FedAvg.
4. The paper provides a mathematical foundation for the convergence and consistency of the Gaussian weights, adding robustness to the proposed method.

**Weaknesses:**

1. Although the algorithm mitigates communication overhead, the interaction matrix and clustering computations may still introduce complexity, especially in large-scale FL deployments with a high number of clients.
2. The paper doesn’t extensively discuss the implications of FedGWC’s clustering on privacy, especially since clustering-based methods might reveal distribution characteristics indirectly.
3. The clustering results are influenced by parameters such as the RBF kernel spread, which could require extensive tuning. This dependency may limit FedGWC’s practicality across different datasets without manual optimization.
4. The empirical validation is limited to specific datasets (CIFAR10, CIFAR100, and FEMNIST). Broader evaluations on diverse datasets, including real-world applications, could strengthen the practical applicability of the findings.

**Questions:**

I would be appreciated if the authors can address my concerns listed in the weaknesses, including the communication efficiency, privacy, scalability, and more comprehensive experiments.

---

> ### Author Response · Authors · 2024-11-23
> **Reply to Reviewer YzY4 (Concerns 1-2)**
>
> **Concern 1: Although the algorithm mitigates communication overhead, the interaction matrix and clustering computations may still introduce complexity, especially in large-scale FL deployments with a high number of clients.**
>
> We sincerely thank the reviewer for their observation. We would like to emphasize that the clustering computations in FedGWC are performed on the server side. We know that computational costs are a bottleneck in federated learning, particularly on client devices with limited resources. For this reason, we offload the clustering computations to the server, while the clients perform the same tasks as in other algorithms, like FedAvg, without additional computational complexity or memory requirements.
>
> In comparison to the clustering baselines, our approach is computationally more efficient, for instance in FeSEM the server computes pairwise distances between model parameters, which are expensive computations in arguably really high dimensional spaces. Notably, our computations involve simple computations (e.g. averaging, or evaluating Gaussian kernels) on the loss values —scalar quantities—rather than handling the model parameters, which are high-dimensional. Additionally, the interaction matrix in our algorithm is updated incrementally, and the increments are highly sparse matrices. This sparsity becomes particularly advantageous in large-scale federated learning deployments, as it keeps memory usage and computational overhead manageable.
>
> We have updated our manuscript to explicitly clarify and emphasize these points by adding in appendix a section addressing qualitatively communication and computational overhead, Appendix D.
>
> We hope this explanation clarifies the practical efficiency of our algorithm and its suitability for large-scale FL scenarios. We remain open to further feedback and are grateful for the reviewer’s support and thoughtful comments.
>
>
> **Concern 2: The paper doesn’t extensively discuss the implications of FedGWC’s clustering on privacy, especially since clustering-based methods might reveal distribution characteristics indirectly.**
>
> We thank the reviewer for raising this important concern regarding the privacy implications of FedGWC’s clustering approach. as privacy is indeed a primary consideration in federated learning.
>
> In the FedGWC framework, clients only transmit their empirical loss vectors $l_k^{t,s}$​ to the server, alongside clients’ models, as in FedAvg.  Note that communicating the empirical losses has been proposed for the first time by Cho et al, 2022. While there might be concerns about potential privacy leakage from sharing this data, it is important to emphasize that the server operates solely on aggregated statistics, rather than individual client data. This ensures that sensitive client information remains private. To further strengthen privacy, the Secure Aggregation protocol (Bonawitz et al, 2016) can be employed. This protocol ensures that only aggregated results are shared, effectively preventing any raw client data from being exposed.
>
> Additionally, FedGWC has been designed to prevent degenerate clusters (i.e., clusters with only one or two clients) to maintain privacy guarantees. Each cluster in our method includes at least three clients, ensuring that no single client’s data can be isolated or inferred. This approach contrasts favorably with baseline methods, where degenerate clusters of single clients can arise, potentially leading to privacy vulnerabilities.
>
> We have updated the manuscript by adding Appendix C: Privacy of FedGWC, where we recall the fact that our algorithm does not involve sharing private information, and could rely on a Secure Aggregation layer. We thank them again for pointing out this critical area for clarification.
>
> *Cho, Yae Jee, Jianyu Wang, and Gauri Joshi. "Towards understanding biased client selection in federated learning." International Conference on Artificial Intelligence and Statistics. PMLR, 2022.
> Bonawitz, Keith, Vladimir Ivanov, Ben Kreuter, Antonio Marcedone, H. Brendan McMahan, Sarvar Patel, Daniel Ramage, Aaron Segal, and Karn Seth. "Practical secure aggregation for federated learning on user-held data." 2016.*
>
> *Bonawitz, Keith, et al. "Practical secure aggregation for federated learning on user-held data." arXiv preprint arXiv:1611.04482 (2016).*

---

> ### Author Response · Authors · 2024-11-23
> **Reply to Reviewer YzY4 (Concerns 3-4)**
>
> **Concern 3: The clustering results are influenced by parameters such as the RBF kernel spread, which could require extensive tuning. This dependency may limit FedGWC’s practicality across different datasets without manual optimization.**
>
> We sincerely thank the reviewer for bringing up this concern regarding the potential sensitivity of FedGWC to its hyper-parameter, the RBF kernel bandwidth $\beta$. We would like to emphasize that $\beta$ is the sole hyper-parameter of FedGWC, and we have conducted a thorough sensitivity analysis across different datasets, presented in Table 6 of Appendix F. The results demonstrate that FedGWC exhibits stable performance across varying values of $\beta$, underscoring its robustness. Specifically, our method achieved the following balanced accuracy scores (mean $\pm$ standard deviation): CIFAR-10 $(74.48 \pm 1.04)$, CIFAR-100 $(51.26 \pm 1.51)$, and FeMNIST $(75.94 \pm 0.18)$.
>
> In contrast, existing baselines, such as IFCA and FeSEM, involve additional hyper-parameters that significantly influence their performance, including the number of clusters, which must be pre-specified. This requirement is impractical in real-world scenarios where the optimal number of clusters is not known a priori. Appendix H (Table 6) provides a detailed comparison, illustrating that changes in the number of clusters for IFCA and FeSEM result in noisier and less predictable performance trends.
>
> Furthermore, the interpretability of $\beta$ adds practical value to FedGWC. The parameter intuitively governs the strength of correlations between clients’ similarity vectors, making it straightforward to understand and adjust. By comparison, the hyper-parameters in other baselines, such as CFL, lack direct interpretability and are prone to degenerative clustering outcomes under slight variations.
>
> We hope this explanation addresses the reviewer’s concern and highlights the practical advantages of FedGWC in terms of hyper-parameter robustness and interpretability. Thank you again for the thoughtful feedback.
>
> **Concern 4: The empirical validation is limited to specific datasets (CIFAR10, CIFAR100, and FEMNIST). Broader evaluations on diverse datasets, including real-world applications, could strengthen the practical applicability of the findings.**
>
> We acknowledge the importance of evaluating FedGWC on a broader range of datasets, including real-world applications, to further demonstrate its practical applicability.
>
> However, our current experiments already demonstrate the effectiveness of FedGWC in handling heterogeneous data distributions. In particular, the CIFAR100 dataset is widely recognized in the literature as a benchmark to evaluate and compare FL algorithms against the issues of heterogeneity . Specifically, our experiments on this dataset demonstrate that FedGWC consistently outperforms baseline methods with respect to balanced accuracy and clustering metrics.
>
>
> To further address the reviewers’ concern, we have updated the manuscript to include a theoretical analysis in Appendix E, outlining the use of maximum-likelihood estimation of the Dirichlet alpha parameter to quantify the heterogeneity of the groups detected by FedGWC.
>
> We are currently running experiments on more realistic datasets and we are committed to expanding our empirical evaluation in the camera-ready version by including the final results, further demonstrating the practical applicability of FedGWC.

---

> > ### Comment · Reviewer_YzY4 · 2024-11-25
> >
> > Thanks for your rebuttal. But I lean to keep my score untill the authors can provide experimental results (even partial results) on more large-scale datasets.

---

> > > ### Author Response · Authors · 2024-12-03
> > > **Official Comment by Authors**
> > >
> > > We thank the reviewer for their feedback. To address this concern, we extended our experimental analysis to include the Office-Home dataset, which presents a more realistic and highly heterogeneous scenario. In this setting, FedGWC, with RBF spread  $\beta = 4$ demonstrated its potential by improving the performance of FedAvg, achieving a balanced accuracy of 81.6\% compared to 81.3\% of FedAvg. Importantly, FedGWC successfully detected two clusters, reflecting its ability to adapt to the diverse distributions present in this dataset. This shows that FedGWC achieves better performances, not only in presence of class imbalance, showing its robustness also in presence of domain heterogeneity. This feature of FedGWC is fundamental in real-world and large-scale FL scenarios, where data distributions across clients, in terms of feature and class imbalance, are often characterized by strong heterogeneity. The detection of two clusters aligns with the expected domain groupings in the Office-Home dataset, which contains multiple visual domains, and confirms that FedGWC effectively captures the underlying structural differences among clients.
> > > These results are promising and highlight the effectiveness of FedGWC in clustering clients according to distributional similarities. For the camera-ready version, we will incorporate a comprehensive comparison with other clustering baselines. Given the high heterogeneity of the Office-Home dataset, we are confident that FedGWC will achieve superior results relative to existing methods.

---

### Official Review · Reviewer_hVEc · 2024-11-04

**Soundness:** 2
**Presentation:** 2
**Contribution:** 2
**Rating:** 5
**Confidence:** 4

**Summary:**

To address the heterogeneity and class imbalance issues in FL, this paper proposes FedGWC, a novel PFL method. Further experiments show that FedGWC outperforms existing FL methods. The clients are grouped into clusters with similar data distribution based on an interaction matrix.

**Strengths:**

- Comprehensive experiments
- mathematical analysis.

**Weaknesses:**

- This paper is hard to follow, especially in the methodology part. e.g, the description of Eq(1). It is also unclear how we get the clusters based on Algorithm 1.
- What’s the meaning of the dash line in Figure1 and how we get the average loss process.
- The relation between $L_k^{t,s}$ and $l_k^{t,s}$.
- What if some clients are hardly sampled, does that affect his reward estimate?
- In the line 259, “relying only on the values of the Gaussian weights is insufficient to identify clusters of clients with similar data distributions, as they do not capture the interactions among pairs of clients.” However, it is not clear why we need the interactions among pairs of clients. I can not catch the motivation.
- What is the clients’ sampling rate at each round?
- It is not applicable that this method needs 20000 communication rounds to be converged.
- Extra communication and computation overhead analysis is needed to compare with baselines.

**Questions:**

See weakness.

---

> ### Author Response · Authors · 2024-11-23
> **Reply to Reviewer hVEc (Concern 1)**
>
> **Concern 1: This paper is hard to follow, especially in the methodology part. e.g, the description of Eq(1). It is also unclear how we get the clusters based on Algorithm 1**
>
> We thank the reviewer for their feedback. In response, we have updated the methodology section to improve clarity and readability. (LL 213-216, LL265-269)
>
> *Explanation for Eq.1: Gaussian Rewards*
>
> The idea behind Equation (1) is the following: for any local iteration $s$, we use an unnormalized Gaussian kernel probability density function (pdf) centered on the mean loss process, with variance equal to the sample variance of the loss process. This allows us to locate how close each client’s loss is to the center of the empirical confidence interval, thus giving a likelihood measure of each client belonging to the same learning process distribution. Indeed, the Gaussian rewards close to 1 are associated with clients whose losses are near the average Gaussian process (with respect to the standard deviation), and the reward decreases as the losses deviate from the mean.
>
> The Gaussian Rewards are scalar values that are an absolute statistical estimator of the expectation of the theoretical reward and are a crucial point motivating the theoretical framework and the setting of FedGWC.
>
> To fully capture distributional similarities among clients, we extend the use of Gaussian rewards by considering not only the individual client losses but also the relationships between clients' learning patterns over training iterations. Specifically, this involves the same mechanism to quantify how closely each client's loss trajectory aligns with those of other clients.
>
> These relationships reflect how data heterogeneity and class distribution affect the evolution of model updates across clients. Considering these inter-client dynamics, we can construct a richer representation of similarity that accounts for both individual performance and collective behavior.
>
> *From Gaussian Weights to Clustering Algorithm*
>
> Extending the formulation of the Gaussian weights to a broader and more general formula, it is possible to naturally define this correlation, or similarities between clients, i.e., the off-diagonal elements of the interaction matrix $ P $,  involved in the clustering algorithm (Alg. 1).– the diagonal elements are indeed the Gaussian Weights.
>
> As far as Alg.1 is concerned, once the interaction matrix $P$ has converged, i.e., the mean squared error between two consecutive steps is below a small threshold, we compute a symmetric feature matrix, which is the affinity matrix $W$. In the paper, we prove the convergence properties of the matrix, which are further validated by the plot in Fig.4, App.H, where we show the norm of the error between two consecutive communication rounds.
>
>  The symmetrization of the interaction matrix $P$ into the affinity matrix $W$ is not only a requirement for spectral clustering but serves to refine the representation of inter-client relationships. By ensuring symmetry, this transformation provides a balanced and reciprocal view of client interactions, mitigating potential biases and emphasizing stable, bidirectional similarities. Moreover, symmetrization reduces noise sensitivity in the interaction matrix and enhances the robustness of the clustering process, allowing spectral clustering to effectively capture the underlying structure of the data. This step ensures that the affinity matrix $W$ encodes meaningful geometric relationships, facilitating the identification of coherent clusters that align with the distributional heterogeneity among clients. Thus, the spectral clustering is computed on the symmetric version of $P$.
>
> To determine the number of clusters, the server performs spectral clustering for a range of cluster numbers $(2, 3, ..., n_{max})$. For each choice of $n$, we check the Davies-Bouldin (DB) score, which quantifies the compactness and separation of clusters. A lower DB score indicates well-separated, compact clusters, while a higher DB score reflects overlapping or poorly defined clusters. The DB score is particularly suitable as it balances intra-cluster variance and inter-cluster distance, making it robust in scenarios with heterogeneous data distributions. If the DB score for all cluster numbers is greater than 1, this indicates that no meaningful clustering occurs, and we keep a single cluster. Otherwise, we select the value of $n$ that gives the best DB score (below 1).
>
> Our approach allows the optimal number of clusters to be determined automatically. Conversely, algorithms like IFCA or FeSEM require the number of clusters to be set as a hyperparameter. This self-adapting feature makes our approach more suitable for real-world scenarios, where the optimal number of clusters is typically unknown a priori.
> We hope this additional clarification addresses the reviewer’s concern and makes the methodology section easier to follow.

---

> ### Author Response · Authors · 2024-11-23
> **Reply to Reviewer hVEc (Concerns 2-4)**
>
> **Concern 2: What’s the meaning of the dash line in Figure1 and how we get the average loss process**
>
> In the left plot of Figure 1, the dashed lines correspond to the empirical losses $ l_k^{t,s} $ of the sampled clients during a fixed communication round $ t $. From these empirical losses, we compute the mean process $ \hat{\mu}^{t,s} = 1/ |\mathcal{P}_t| \sum_kl_k^{t,s} $ (depicted as the bold black curve) and the standard deviation $\( \hat{\sigma}^{t,s} \)$, which define the confidence interval $\hat{\mu}^{t,s} \pm \hat{\sigma}^{t,s} $ (shaded in light blue in the plot). These quantities $ l_k^{t,s} $ are used for deriving the average process.
>
> The derivation of these metrics is detailed in Equation 1, which also outlines the computation of the Gaussian weights and their role in the framework.
>
> In particular, if the process $ l_k^{t,s} $ consistently lies within the confidence interval $ \hat{\mu}^{t,s} \pm \hat{\sigma}^{t,s} $ across all local iterations $ s = 1, \dots, S$, the client is more likely to belong to the same underlying distribution as the majority of sampled clients, leading to a higher empirical reward.
>
> To address this concern, we have revised the caption of Figure 1 to clarify the computation of the average process and standard deviation and their significance in our methodology.
>
> **Concern 3: The relation between L_k^{t,s} and l_k^{t,s}**
>
> We thank the reviewer for their observation and appreciate the opportunity to clarify this point. As stated in Section 3.1 (LL 187-188), we use capital letters (e.g., $X$) to denote random variables and lowercase letters (e.g., $x$) to represent their specific observations. This distinction is needed as our theoretical analysis leverages tools from probability theory to establish the convergence of the Gaussian weights and the interaction matrix.
>
> Specifically, $ l_k^{t,s} $ refers to the empirical loss computed on client $ k $, sampled at the $ t $-th communication round during the $s $-th local iteration. Separating the random variable $ L_k^{t,s} $ from its observation $l_k^{t,s} $ is essential, especially in the context of stochastic optimization algorithms. Because of stochastic optimizers, the learning process is inherently stochastic and can be modeled as a random process composed of random variables $ L_k^{t,s} $.
>
>
> We hope this explanation clarifies the notation and its significance in our work.
>
> **Concern4: What if some clients are hardly sampled, does that affect his reward estimate?**
>
> We thank the reviewer for the question. We design FedGWC to mitigate biases in the reward estimate through two key features:
>
> 1. **Unbiased Client Sampling:**  Rather than employing a biased client selection policy, we adopt uniform random sampling to simulate the stochastic availability of clients in a cross-device FL scenario, a common practice in the literature. This ensures that each client is sampled approximately the same number of times over the long run, leading to an empirical sampling frequency that is nearly uniform across the federation. Additionally, to address scenarios where certain clients are not sampled for an extended number of rounds, we incorporate a mechanism that dynamically increases the probability of selecting these clients in subsequent rounds. This approach helps simulate the natural tendency for rarely sampled clients to become available, thus reducing the impact of prolonged under-sampling.
> 2. **Reward Stability for Non-Sampled Clients:** When a client is not sampled in a round, its weight and contribution to the reward estimate remain unchanged. This design choice further reduces potential biases in the reward estimation process, ensuring that unselected clients do not adversely affect the overall calculations.
>
> We have revised the original manuscript to implicitly discuss how these mechanisms were incorporated into the algorithm to limit biases (LL 238- 242). We hope this explanation clarifies the measures taken to address this concern.

---

> ### Author Response · Authors · 2024-11-23
> **Reply to Reviewer hVEc (Concern 5-6)**
>
> **Concern 5: In the line 259, “relying only on the values of the Gaussian weights is insufficient to identify clusters of clients with similar data distributions, as they do not capture the interactions among pairs of clients.” However, it is not clear why we need the interactions among pairs of clients. I can not catch the motivation.**
>
> We thank the reviewer for their question. We provide further explanation of our motivations. The core idea behind our approach is that to effectively capture the interactions between clients’ data distributions, we need to go beyond the Gaussian weights, which are scalar values that provide an absolute measure of how well a client’s data distribution aligns with the global one. While Gaussian weights reflect the fitness of each individual client’s distribution, they do not account for the relationships between clients’ distributions, which is crucial for accurate clustering.
>
> To better capture these interactions, we extend the idea of Gaussian weights to consider the relative similarity between clients’ distributions. By incorporating pairwise interactions, we obtain a more informative measure that helps identify clusters of clients with similar learning patterns. This approach allows for a more refined and effective representation of the clustering process, improving the overall quality of the personalized models. The weights and the interaction matrix are designed to keep track of rewards over time. Furthermore, empirical losses are samples from a random process, so we use this time averaging to obtain a more robust and unbiased estimation of the theoretical rewards.
>
>  This temporal aspect of the interaction matrix captures not only the current distributional similarity but also the evolving correlations between clients across rounds. As such, the matrix quantifies both the interactions and similarities, which are interpretable as temporal correlations, thus providing a more expressive view of the clustering process. This approach ultimately leads to a more refined and effective representation of the clustering, improving the overall quality of the identified groups.
> We have updated the manuscript (LL 265-268) to motivate this point further and provide a more detailed explanation to clarify this crucial aspect of our algorithm.
>
> **Concern6: What is the clients’ sampling rate at each round?**
>
> The sampling rate is set at 10\% for each cluster in the federation, which is a reasonable assumption based on typical values used in the federated learning literature [some citation here]. In heterogeneous scenarios like those in the experimental section, the overall rate across the federation remains approximately 10\%, though it is important to note that this rate can vary depending on the specific client availability and other system conditions.
>
> For smaller clusters, to ensure privacy preservation, we limit the number of participating clients to at least 3. This ensures that the server only has access to aggregated values, preserving the privacy of individual clients within the cluster. Additionally, to avoid exposing sensitive information, we ensure that no cluster contains fewer than 3 clients, preventing the risk of revealing private information from single clients. To further prevent excessive fragmentation and oversampling, we also enforce a minimum cluster size constraint.
>
> We have uploaded the manuscript by adding Theorem A.3, alongside the proof,  in Appendix A, which provides a sufficient condition that guarantees that the sample rate remains unchanged during the training process

---

> ### Author Response · Authors · 2024-11-23
> **Reply to Reviewer hVEc (Concerns 7-8)**
>
> **Concern 7: It is not applicable that this method needs 20000 communication rounds to be converged.**
>
>
> We thank the reviewer for their comment. Our experimental setting aligns with Hsu et al. (ECCV 2020, Appendix A1), where 20k iterations were used for training on CIFAR-100. It has also been established that heterogeneous scenarios require a high number of communication rounds to achieve convergence; for example, Caldarola et al. (ECCV 2022, Fig. 3) reported over 20k rounds in such settings and show that extreme heterogeneous settings require up to 10x more rounds. We used this insight for a fair comparison and let our algorithm and the baselines train for enough iterations.
>
> We would also like to highlight Figure 2 in our manuscript, where we compare FedGWC against other clustering baselines and FedAvg aggregation. This figure demonstrates that FedGWC converges faster than the baselines. While all methods exhibit similar learning trends, our algorithm achieves convergence more efficiently. We have updated the caption of Figure 2 to make this observation clearer.
> We hope this additional context addresses the reviewer’s concern and provides a better understanding of our approach.
>
>
> *Hsu, Tzu-Ming Harry, Hang Qi, and Matthew Brown. "Federated visual classification with real-world data distribution." Computer Vision–ECCV 2020: 16th European Conference, Glasgow, UK, August 23–28, 2020, Proceedings, Part X 16. Springer International Publishing, 2020.*
>
>
> *Caldarola, Debora, Barbara Caputo, and Marco Ciccone. "Improving generalization in federated learning by seeking flat minima." European Conference on Computer Vision. Cham: Springer Nature Switzerland, 2022*
>
>
>
> **Concern 8: Extra communication and computation overhead analysis is needed to compare with baselines.**
>
> We thank the reviewer for their suggestion. We are aware that communication and computation costs are crucial aspects of federated learning, and we appreciate the opportunity to clarify this point.
>
> Our method does not introduce additional computation on the client side. Clients are only required to communicate their local models, as is standard in any FL algorithm (McMahan et al, NIPS 2016), along with the empirical losses at round $ t $, which only necessitate negligible additional communication costs. These losses are represented as a vector with $ S $components, where $ S $ corresponds to the number of local iterations (i.e., the product of local epochs and the number of batches). In our setup, $S = 8 $.
>
> Since $S  \ll |\Theta| $, where $|\Theta| $ denotes the size of the model's parameter space, the communication of these loss values is negligible in comparison to the transmission of model weights. Furthermore, all clustering operations and weight computations are performed on the server side, ensuring that the computational overhead on the client side remains unchanged.
>
> As a result, the overall communication and computational overhead on the client side is equivalent to that of the chosen FL method (e.g., FedAvg). We believe that our method meets the fundamental requirements of federated learning, namely, minimizing costs while maintaining high performance without disclosing clients’ private information. In fact, the cost associated with our method is lower than that of the benchmark methods, although we acknowledge that further detailed comparisons could provide additional insights.
>
> We have updated our manuscript to explicitly clarify and emphasize these points by adding in appendix a section addressing qualitatively communication and computational overhead, Appendix D. We hope this addresses the reviewer’s concern and provides sufficient clarity.
>
>
> *McMahan, H. Brendan, et al. "Federated learning: Strategies for improving communication efficiency." Proceedings of the 29th Conference on Neural Information Processing Systems (NIPS), Barcelona, Spain. 2016.*

---

> > ### Comment · Reviewer_hVEc · 2024-11-27
> >
> > Thanks for your careful response. However, there are lots of works that use ResNet-9/18 to conduct their experiments which does not need many communication rounds. Is this method still work when some FL scenario use big model which also reduces the communication rounds. Whether the fewer communication round reduce the efficacy of the clustering method, e.g., only 100 communication rounds.

---

> > > ### Author Response · Authors · 2024-11-27
> > >
> > > Could the reviewer provide references for these works? We are happy to address the question, but without precise references, it is not possible to assess whether the mentioned settings are compatible with our algorithm.

---

> > > > ### Comment · Reviewer_hVEc · 2024-11-27
> > > >
> > > > Shi, Yujun, et al. "Towards Understanding and Mitigating Dimensional Collapse in Heterogeneous Federated Learning." The Eleventh International Conference on Learning Representations.
> > > >
> > > > Jhunjhunwala, Divyansh, Shiqiang Wang, and Gauri Joshi. "FedExP: Speeding up Federated Averaging via Extrapolation." International Conference on Learning Representations. 2023.
> > > >
> > > > Mendieta, Matias, et al. "Local learning matters: Rethinking data heterogeneity in federated learning." Proceedings of the IEEE/CVF Conference on Computer Vision and Pattern Recognition. 2022.
> > > >
> > > > Tang, Zhenheng, et al. "Virtual homogeneity learning: Defending against data heterogeneity in federated learning." International Conference on Machine Learning. PMLR, 2022.
> > > >
> > > > Song, Congzheng, Filip Granqvist, and Kunal Talwar. "Flair: Federated learning annotated image repository." Advances in Neural Information Processing Systems 35 (2022): 37792-37805.
> > > >
> > > > Son, Ha Min, et al. "FedUV: Uniformity and Variance for Heterogeneous Federated Learning." Proceedings of the IEEE/CVF Conference on Computer Vision and Pattern Recognition. 2024.
> > > >
> > > > Guo, Yaming, et al. "Out-of-distribution generalization of federated learning via implicit invariant relationships." International Conference on Machine Learning. PMLR, 2023.
> > > >
> > > > Yang, Zhiqin, et al. "FedFed: Feature distillation against data heterogeneity in federated learning." Advances in Neural Information Processing Systems 36 (2024).
> > > >
> > > > Qu, Zhe, et al. "Generalized federated learning via sharpness aware minimization." International conference on machine learning. PMLR, 2022.
> > > >
> > > > There are plenty of papers use Resnet-18 to conduct experiments, and also they don't need 20000 communication rounds.

---

> ### Author Response · Authors · 2024-12-03
> **Official Comment by Authors**
>
> We thank the reviewer for their observation. We emphasize that our study primarily focuses on settings akin to cross-device scenarios, which are particularly characterized by high heterogeneity (Dirichlet’s parameter $\alpha = 0.05$), low participation rates, and limited availability. Most prior works rely on full participation when dealing with a smaller number of clients, Mendieta, Matias, et al (2022), Shi et al., 2022. In cases involving a larger number of clients, they typically employ a participation rate of 20% (e.g., Jhunjhunwala et al., 2023; Shi et al., 2022), which is double compared to our choice of 10%.
>
> To address this, we replicated the settings of Jhunjhunwala et al. (2023). Specifically, we used a participation rate of $\rho = 0.2$, $S = 20$ local iterations, and the same number of communication rounds as reported in their work: 400 for CIFAR-10 and 500 for CIFAR-100. We maintained the heterogeneity level consistent with our original experiments to simulate a more realistic heterogeneous scenario.
>
> Additionally, we tested multiple values for the hyperparameter $\beta \in (0.1, 0.5, 1, 2, 4)$, which controls the spread of the RBF kernel used to compute the similarity matrix. The results were compared with FedAvg under these settings, using ResNet18, and MobileNetV2. In both scenarios, FedGWC demonstrated faster convergence and achieved superior performance in cases where early clustering was detected. The results are summarized in the following tables.
> On CIFAR-10, in the configuration where clustering was detected, we surpassed FedAvg by 6%. On CIFAR-100, where heterogeneity poses a greater challenge, we achieved approximately $30$ % balanced accuracy with clustering, compared to $13$% with FedAvg. In both scenarios, FedGWC demonstrated faster convergence and superior performance in cases where early clustering was detected. In the following tables we present the results of this ablation.
>
> Table 1. Experiments on heterogeneous Cifar10, and 400 communication rounds with MobileNetV2 architecture
>
> | Method                         | Number of clusters | Balanced Accuracy |
> |--------------------------------|--------------------|-------------------|
> | FedAvg                         | 1                  | 45.8 %            |   |   |
> | FedAvg + FedGWC ($\beta = .1$) | 1                  | 45.8 %            |
> | FedAvg + FedGWC ($\beta = .5$) | 1                  | 45.8 %            |
> | **FedAvg + FedGWC ($\beta = 2$)**  | **5**                  | **52.8 %**            |
> | FedAvg + FedGWC ($\beta = 4$)  | 3                  | 51.6%             |
>
> Table 2. Experiments on heterogeneous Cifar100, and 500 communication rounds with MobileNetV2 architecture
> | Method                         | Number of clusters | Balanced Accuracy |
> |--------------------------------|--------------------|-------------------|
> | FedAvg                         | 1                  | 13.4 %            |
> | FedAvg + FedGWC ($\beta = .1$) | 1                  | 13.4 %            |
> | FedAvg + FedGWC ($\beta = .5$) | 1                  | 13.4%             |
> | FedAvg + FedGWC ($\beta = 2$)  | 1                  | 13.4%             |
> | **FedAvg + FedGWC ($\beta = 4$**)  | **5**                  |**30.4%**             |
>
> Table 3. Experiments on heterogeneous Cifar10, and 400 communication rounds with ResNet18 architecture
> | Method                         | Number of clusters | Balanced Accuracy |
> |--------------------------------|--------------------|-------------------|
> | FedAvg                         |  1                 | 47.5%             |
> | FedAvg + FedGWC ($\beta = .1$) | 1                  | 47.5%             |
> | FedAvg + FedGWC ($\beta = .5$) | 1                  | 47.5%             |
> |**FedAvg + FedGWC ($\beta = 2$)**  | **2**                  | **50.6%**             |
> | FedAvg + FedGWC ($\beta = 4$)  | 1                  | 47.5%             |
>
>
> Table 4. Experiments on heterogeneous Cifar100, and 500 communication rounds with ResNet18 architecture
>
> | Method                         | Number of clusters | Balanced Accuracy |
> |--------------------------------|--------------------|-------------------|
> | FedAvg                         | 1                  | 13.5%             |
> | FedAvg + FedGWC ($\beta = .1$) | 1                  | 13.5%             |
> | FedAvg + FedGWC ($\beta = .5$) | 1                  | 13.5%             |
> | FedAvg + FedGWC ($\beta = 2$)  | 2                  | 18.5%             |
> | **FedAvg + FedGWC ($\beta = 4$)**  | **4**                  |**32.5%**            |

---

### Author Response · Authors · 2024-11-23

We thank the Reviewers for their comments, which helped us improve the clarity of our contributions.

The most frequent concerns raised during the review process are related to the computational cost and communication overhead of the algorithm, privacy preservation, clarifications on the method, and the participation rate. To address these, we have updated the manuscript (with changes highlighted in blue) as follows:


- Communication and Computational Costs: We added Appendix D to demonstrate that FedGWC does not increase communication overhead or computational costs. Clients only communicate model weights and empirical loss—negligible compared to model parameters—and all clustering computations are handled by the server.

- Privacy Considerations: In Appendix C, we included a detailed explanation showing that FedGWC adheres to the privacy paradigms of federated learning.

- Clarifications on the Method: We revised captions, tables, and descriptions throughout the manuscript to improve clarity. We also expanded the discussion of clustering metrics (Silhouette, Davies-Bouldin, and Rand Index) to justify their selection, Appendix E.

- Participation Rate: We strengthened the theoretical section by adding a new theorem that establishes a sufficient condition for conserving the overall participation rate during clustering (Theorem A.3).

We believe these updates address all concerns raised by the reviewers and that the revised manuscript significantly enhances the clarity and quality of our work.

Sincerely,

The authors of submission 9633

---

### Comment · Area_Chair_RebS · 2024-11-25
**Acknowledge the author responses**

Dear Reviewers,

Thank you very much for your effort. As the discussion period is coming to an end, please acknowledge the author responses and adjust the rating if necessary.

Sincerely,
AC

---

### Comment · Area_Chair_RebS · 2024-11-28
**Discussion needed**

Dear Reviewers,

As you are aware, the discussion period has been extended until December 2. Therefore, I strongly urge you to participate in the discussion as soon as possible if you have not yet had the opportunity to read the authors' response and engage in a discussion with them. Thank you very much.

Sincerely,
Area Chair

---

### Meta-Review · Area_Chair_RebS · 2024-12-19

**Metareview:**

This paper proposes a personalized federated learning method, which groups clients based on their data distribution.  Such clustering allows training of a more robust and personalized model on the identified clusters.  Although the reviewers found that the paper is interesting, they also raised several concerns on correctness and evaluation of the proposed method.  In particular, one reviewer pointed out the scalability issue for a large-scale dataset.  No reviewer was enthusiastic about the acceptance of this paper during the discussion period.  Thus, based on the reviewers' opinions, I recommend a reject.

**Additional Comments On Reviewer Discussion:**

Reviewers hVEc and Wepx increased their rating to 5 due to the additional results.  However, all the reviewers were reluctant to increase the rating above 6 mainly due to the scalability issue and the unclear connection of clustering to the accuracy.

---

### Decision · Program_Chairs · 2025-01-22

Reject